**METHOD**

# *aurora*: a machine learning gwas tool for analyzing microbial habitat adaptation

Dalimil Bujdoš[1,2], Jens Walter[1,2,3] and Paul W. O'Toole[1,2*]

*Correspondence:
pwotoole@ucc.ie

[1] APC Microbiome Ireland,
University College Cork, National
University of Ireland, Cork, Ireland
[2] School of Microbiology,
University College Cork, National
University of Ireland, Cork, Ireland
[3] Department of Medicine,
University College Cork, National
University of Ireland, Cork, Ireland

## Abstract

A primary goal of microbial genome-wide association studies is identifying genomic variants associated with a particular habitat. Existing tools fail to identify known causal variants if the analyzed trait shaped the phylogeny. Furthermore, due to inclusion of allochthonous strains or metadata errors, the stated sources of strains in public databases are often incorrect, and strains may not be adapted to the habitat from which they were isolated. We describe a new tool, *aurora*, that identifies autochthonous strains and the genes associated with habitats while acknowledging the potential role of the habitat adaptation trait in shaping phylogeny.

**Keywords:** GWAS, Microbial GWAS, Habitat adaptation, Machine learning, *Limosilactobacillus reuteri*, *Lactiplantibacillus plantarum*, *Salmonella* Typhimurium, *Mycobacterium paratuberculosis*, Autochthonous, Allochthonous

## Background

GWAS, or genome-wide association study, is an approach for identifying genetic variations (including genes, SNPs, *k*-mers) that are associated with complex traits (*syn.* phenotypes). GWAS is highly effective for identifying genetic loci associated with a wide range of complex phenotypes [1–3]. Human GWAS studies have successfully identified genomic elements associated with height, obesity, and many diseases such as Alzheimer's and diabetes [3, 4]. The development of microbial GWAS (mGWAS) has lagged behind that of human GWAS. While microbial genomes are cheaper to sequence, their high plasticity and fast evolution, combined with frequent horizontal gene transfer, has made mGWAS difficult to apply successfully [1, 2]. Additionally, microbes reproduce asexually which may lead to the formation of lineages that are genetically similar or even clonal [5, 6]. This results in a rapidly fluctuating clonal structure in which genomic variants can be in genome-wide linkage disequilibrium (LD), making it difficult to identify the loci causally responsible for a certain phenotype. LD is especially strong in microbes where clonal expansion is evident such as in causative agents of infectious diseases [7, 8]. Recombination or gene loss disrupts LD thus alleviating the cost to fitness associated

with carrying non-causal variants that are co-inherited with the causal variants. Thus, LD is less confounding in diverse datasets where samples were gathered over a long period from multiple geographical locations. A successful mGWAS tool needs to be able to account for both LD and complex genetic relatedness, within microbial populations.

Human GWAS analysis focuses primarily on the association of traits with SNPs, but mGWAS offers more options. These include SNPs, genes, *k*-mers, or unitigs (overlapping set of *k*-mers) as the genetic variants [1], and in practice, it is possible to test all of these variants depending on the nature of the analyzed trait. The presence/absence of genes is the most interpretable approach but may result in omitting important causal sequence variants. On the other hand, downstream processing of significantly associated *k*-mers or SNPs can be challenging since there can be thousands of significant hits. Traditional methods like mapping the significant variants to a reference genome are not possible if the analyzed species are genomically very diverse. Recently, it was suggested to use unitigs instead of *k*-mers as the tested variant [9, 10]. Because analysis of all genomic variants offers distinct advantages, novel bioinformatic tools should be designed to work with any feature (presence/absence of genes, unitigs, SNPs, even whole pathways, and metabolic modules).

The currently available mGWAS tools are not suitable for identifying genetic variants responsible for habitat adaptation because they discard, or disadvantage variants that are associated with phylogenetic lineages. Genetic variations whose presence strongly depend on the evolutionary history are commonly referred to as "lineage effects" while variations influencing phenotype independently of the phylogenetic background are referred to as "locus effects" [11, 12]. Although complex traits are often encoded by both types [11, 13], most currently available mGWAS tools only focus on locus effects. For example, the mGWAS tool Hogwash [14] removes all genetic elements that were not gained or lost at least twice in the phylogeny. The Seer algorithm [15] cannot analyze phenotypes that are uniquely associated with just one lineage. However, if a species has a restricted habitat range over long evolutionary time, phylogenetic lineages can emerge that are specific to this habitat, which might apply to host specific symbionts [16–19]. Thus, the habitat specificity is a phenotype well reflected by both locus effects and phylogenetic structure. By strictly accounting for the population structure, the currently available mGWAS tools fail to identify adaptive variants that influenced bacterial diversification, because they assume that causal variants had a negligible effect on the analyzed phenotype. This assumption can be safely applied in human GWAS because complex human traits are mostly under low selection pressure [20]. Some currently used mGWAS tools were inspired by tools used in human GWAS. For example, Pyseer [9, 21] and DBGWAS [10] use linear models with principal components to account for the population structure. Another group of mGWAS tools uses a phylogenetic tree to calculate the number of tree branches or nodes where the phenotype and genotype interacted [14, 22, 23]. These tools have been successfully used to identify causal markers of antimicrobial resistance, virulence, and invasiveness [24–27], which can be the result of short-term adaptation (encoded by locus effects) that tend not to influence the long-term phylogeny and form lineages. The problem of causal variants concordant with phylogeny had already been recognized. Earle et al. (2016) proposed a method named *bugwas* to identify causal lineage effects [11]. However, this method has the limitation

that causal locus and lineage effects are identified in two separate analyses which does not allow a direct comparison of their significance. The *bugwas* method is also suitable mostly for identification of variants that arose just once during the course of evolution. Variants strongly correlated with phylogeny but present in multiple lineages with multiple corresponding occurrences in the phylogeny are difficult to identify using *bugwas*. Since dividing causal variants into locus and lineage effects is an artificial partitioning, combining analysis of the two in one tool would be a significant improvement. It is also desirable to have a robust tool that does not make an assumption about the effect of the analyzed trait on the phylogeny, while it still accounts for the clonal structure of the microbial population.

Constructing a dataset is a critical step in mGWAS. The accuracy of the mGWAS analysis is highly dependent on the quality of metadata associated with the samples. Metadata errors can arise from various sources, such as incorrect taxonomic identification, faulty measurement of the trait, and mistakes in recording strain provenance. Since mGWAS often analyses thousands of strains, some errors are bound to occur [28]. Moreover, the source of isolation of a strain is not necessarily the environment to which it is adapted, as strains might be allochthonous (not formed where found). Metadata errors and strain allochthony can significantly affect the accuracy and validity of mGWAS results because mGWAS tools use strict adjustment for population structure which gives large weight to outliers (possible erroneously labeled or allochthonous strains). Moreover, there may be strains or entire lineages that are well adapted to a habitat while other lineages of the same species do not have a strict habitat preference [29–31]. This makes for an exceptionally difficult dataset as the host-adapted strains need to be analyzed separate from the strains with a broad host range. A new generation of mGWAS tools thus needs to be able to identify strains that possess the causal genomic variants prior to the mGWAS analysis. Lastly, some phenotypes are not heritable and in such a case mGWAS tools should not generate false positives Thus, it is desirable to have a tool that can establish if the entire species displays any genomic variants associated with the recorded trait.

Here we present a new tool called *aurora* (AUtochthonous, Random fOrest, Random wAlk) that can deal with the confounders described above. *aurora* was thoroughly benchmarked against the most commonly used mGWAS tools that can use the pangenome matrix as a feature input table. Multiple simulated datasets and real datasets were used to show that *aurora* can correctly analyze non-heritable phenotypes and is the only tool that can retrieve the causal genetic variants despite the presence of numerous mislabeled strains and despite the collinearity of the analyzed phenotype and the phylogeny. The *aurora* algorithm is implemented as an R package and is freely available.

## Results

### Functionality of *aurora*

*aurora* has two main functions aurora_pheno() and aurora_GWAS(). The workflow of both functions is depicted in Fig. 1. The parameters of both functions are discussed in detail in Additional file 1. The file also contains an example analysis of the *L. reuteri* dataset. The purpose of function aurora_pheno() is to identify strains that do not possess causal variants associated with the phenotype of interest (i.e., virulence, host

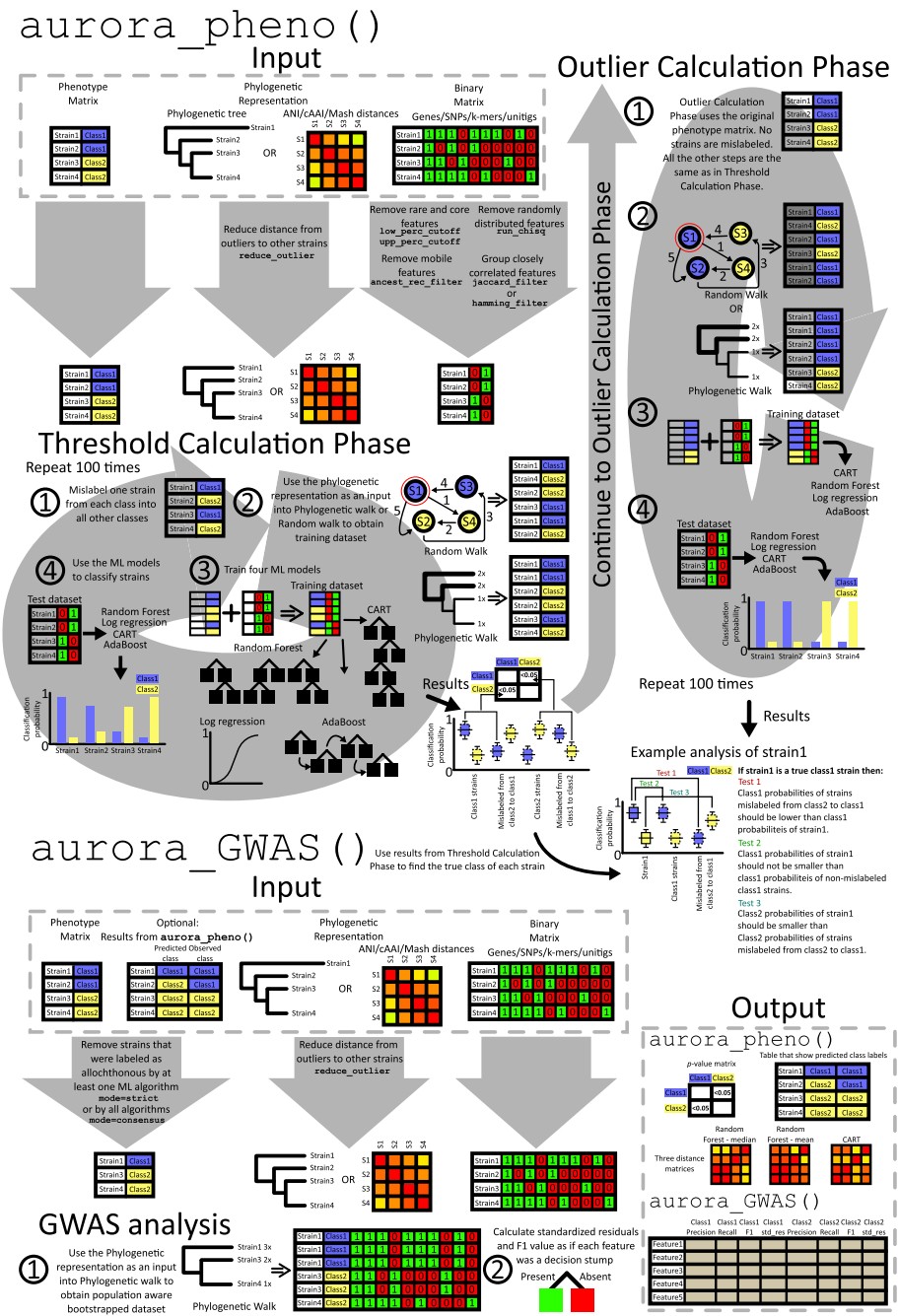

**Fig. 1** Workflow of the *aurora* package

specificity). When host adaptation is analyzed, we refer to strains that are able to stably colonize as autochthonous (formed where found), while strains that are temporarily present but introduced from another habitat are termed allochthonous. A broader term is "mislabeled" which refers to any strain that aurora_pheno() identified as not associated with the observed class. Therefore, mislabeled strains are either allochthonous strains or strains that were erroneously labeled in the user-supplied dataset. These strains should be removed from subsequent mGWAS analysis to preserve maximal power for

identifying causal variants. The first step of aurora_pheno() is to filter the input feature matrix and collapse highly correlated features into a single representative feature. This data matrix then enters the Threshold Calculation Phase. In this phase, a cycle of intentional random strain mislabeling, and subsequent training of four ML models (random forest, AdaBoost, logistic regression, and CART) is repeated. The Threshold Calculation Phase generates threshold distributions composed of classification probabilities obtained from the four ML models. To identify the mislabeled strains these distributions are compared to classification probability distributions obtained in the next step: Outlier Calculation Phase. The Threshold Calculation Phase also outputs *p*-value matrices (visualized in Additional file 1: Fig. S1 for *Limosilactobacillus reuteri* and Additional file 1: Fig. S2 for *Lactiplantibacillus plantarum*) that show whether the analyzed species have variants associated with the classes of the phenotype. The purpose of function aurora_GWAS() is to identify causal features of the investigated phenotype. If results from aurora_pheno() are available, then the strains identified by aurora_pheno()as mislabeled are first removed. aurora_GWAS() calculates simple genotype–phenotype association scores (F1 values and standardized residuals) on a bootstrapped dataset that is adjusted for strain non-independence.

### *aurora* can identify causal variants in simulated data despite inclusion of incorrectly labeled strains

To evaluate the performance of *aurora*, we first created simulated GWAS test data generated by four methods: two multiple state speciation and extinction models (MuSSE1 and MuSSE2) [32], Simurg [33], and a script Simulate_pan_genome.py (Scoary script) build to test a mGWAS tool Scoary [22]. These datasets were designed to represent four different phenotype adaptation scenarios (see Fig. 2). The script Simulate_pan_genome.py creates a dataset with just one causal gene. This scenario was simulated twice: high penetrance—0.85 and medium penetrance—0.6. Both phenotypes were dispersed around the phylogenetic tree indicating an insignificant role of phylogenetic history for this phenotype. This is measured by D value—a measure of phylogenetic signal strength in binary phenotypes [34]. The D values for high and medium penetrance phenotypes were 0.4 and 0.2 respectively. In such scenarios, the causal genes will be repeatedly acquired and lost in the phylogeny which is typical for highly transient phenotypes like virulence and antimicrobial resistance [25–27].

Simurg produces only the pangenome matrix and the phylogenetic tree but not trait distribution. Thus, the phylogenetic tree was split into two lineages, and each was assigned a different class. Such label distribution is typical for host-adapted species where deep branching lineages are associated with a specific host [16–18]. The D value for this dataset was estimated to be $-0.37$ indicating a strong phylogenetic signal for this phenotype. The MuSSE1 model is a more realistic depiction of host adaptation. Each time a strain gains a new causal gene, its speciation rate is increased. This reflects the fitness benefit that the strain acquired with the causal gene. Since this type of model tends to generate causal variants that are lineage effects [11] which some convergence-based models and some linear models cannot identify, a gene distribution where the causal genes are not present exclusively in one lineage was selected (Fig. 2B). The D value for this phylogeny and trait distribution was $-0.53$ indicating a strong influence

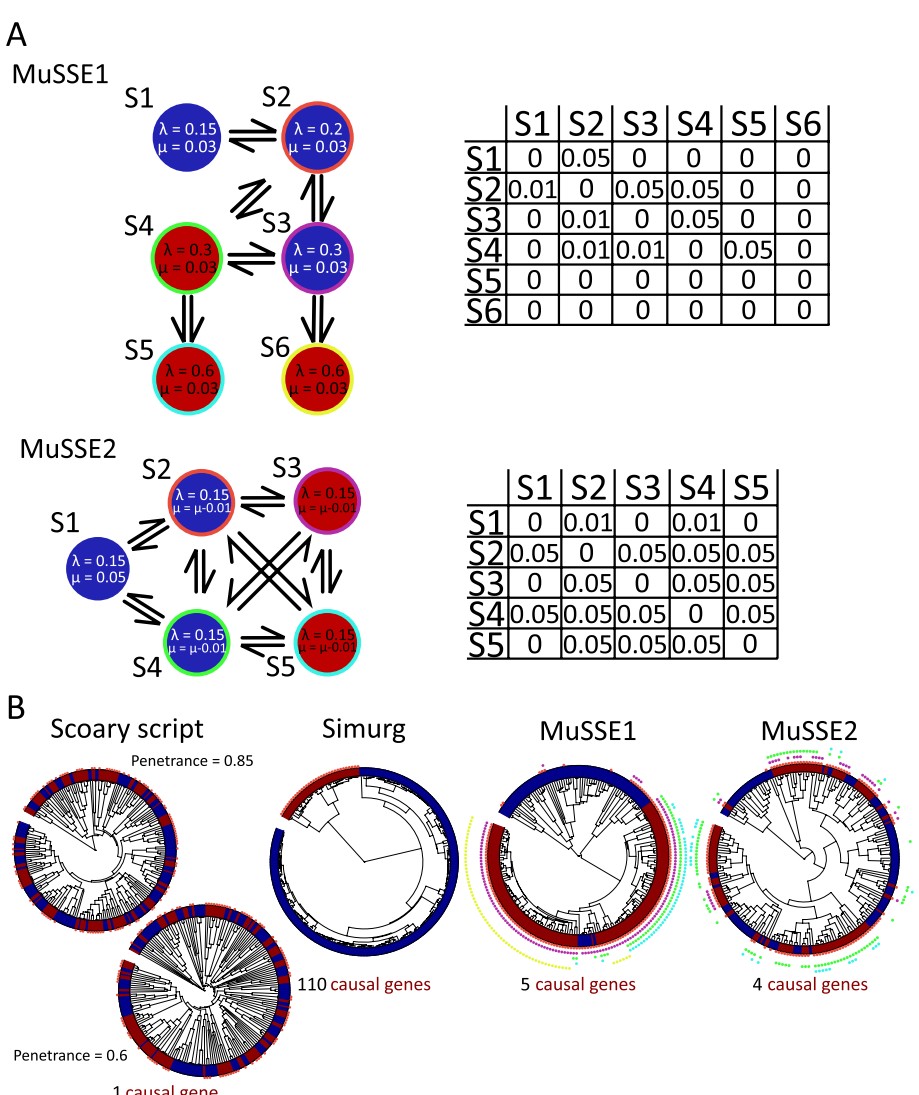

**Fig. 2** Construction and visualization of simulated data. Four simulated datasets were constructed using methods with different assumptions: Simulate_pan_genome.py script published along with Scoary [22], pangenome simulation tool Simurg [33], and two multiple state speciation and extinction models (MuSSE). **A** Visualization of the construction of the two MuSSE models. The speciation (λ) and extinction (μ) rates are shown for each state. The corresponding transition matrices are shown next to the model scheme. In the case of MuSSE1 model, if a strain finished the simulation in a state with blue color (S1, S2, S3) then the strain was considered to belong to the blue class of the phenotype. If the final state was red (S4, S5, and S6), then the strain belongs to the red class of the phenotype. In the case of MuSSE2 model, once a strain passed the red states (S3 and S5), it was considered to belong to the red phenotype even if it later returned to one of the blue states (S1, S2, S4). Each time a strain transitions into a new state, it gains the causal gene that corresponds to the colored border ring. The colored rings around the states correspond to the colored circles in the phylogenetic trees below. Additionally, in MuSSE2 simulation if a strain gains a causal gene (S2, S3, S4, S5) then its current extinction rate is reduced by 0.01. **B** Phylogenetic trees showing how the phenotype classes (inner color strip) and causal genes (outer ring with circles) were distributed. Causal genes are those which contribute to the red phenotype. The subsequent GWAS analysis was used to discover adaptation factors to the red phenotype

of phylogenetic history on the trait. The MuSSE2 model on the other hand is a simulation of a more transient phenotype (D = −0.06). Each time a strain gains a causal gene, its extinction rate decreases. This reflects the increased resistance to a hypothetical inhibitor.

To test how the performance of *aurora* was influenced by mislabeled strains, we intentionally incorrectly labeled 0, 5, 10, and 20% of strains into the opposite class in the simulated data and compared the ability of *aurora* to identify the causal features with that of four popular mGWAS tools that can use a pangenome matrix as an input—Hogwash [14], Scoary [22], Pyseer [21], and TreeWAS [23]. The measure of how successful a particular tool was in identifying the causal genes was the rank of these causal genes in the output. All genes were first ordered by their *p*-values in ascending order and then rank was computed for the causal genes. An average of those ranks was calculated as the final measure of success. In the case of elastic net, the results were sorted based on unadjusted *p*-values. The same procedure was applied to the *aurora* results, but the measure of success was the average rank of features ordered in descending order based on either F1 values (Simurg and the high penetrance Scoary script data) or standardized residuals (all other datasets). If a causal gene was not present in the output, a rank value of 10,000 was assigned to that gene. It was assumed that if a tool did not report a causal variant, the variant is at the end of the list with a *p*-value equal to 1. The number 10,000 was chosen because it reflects the typical pangenome size of a medium-sized dataset (∼500 strains). The mean rank measure is especially suitable if the output of a certain tool has a large number of significant ($p < 0.05$) genes. In such a case, the causal genes would be significant but other genes (false positives) may have even lower *p*-values. The results are shown in Fig. 3.

Apart from the two tests implemented in the Hogwash package, all other tools performed well on the high penetrance dataset produced by the Scoary script. However, only *aurora* and the three Pyseer models predicted rank 1 for the causal gene irrespective of how many strains were incorrectly labeled. It should be noted that the *p*-value predicted by the fixed effects model increased from $9.13 \times 10^{-23}$ to $4.28 \times 10^{-9}$ when 20% of all strains had swapped labels. A similar trend was observed in the case of the linear mixed model. On the other hand, the F1 values produced by *aurora* slightly rose from 0.8 to 0.9. The medium penetrance dataset revealed more about the performance of each tool. Only three methods—*aurora*, fixed effects, and linear mixed effect model, managed to predict low rank when no strains were incorrectly labeled. Importantly, the performance of the said models deteriorated as more strains were given swapped labels while the performance of *aurora* decreased only slightly Fig. 3A.

The Simurg simulation produced a phylogeny with two distinct lineages. There were 110 genes present in the first lineage (red color in Fig. 2B) and absent in all strains in the second lineage. These genes were considered causal. The best mean rank was thus 55, a result which was achieved only by *aurora* and the Subsequent test when no strains were incorrectly labeled (Fig. 3A). Hogwash, Scoary, fixed effects, and linear mixed model were unable to analyze the phenotype despite its simplicity. Only lineage-specific genes are causal for this trait, and it is thus expected that these tools (geared to the analysis of locus effects) would not be able to identify its causal variants. This simulation again showed that *aurora* can identify the incorrectly labeled strains as the tool's performance

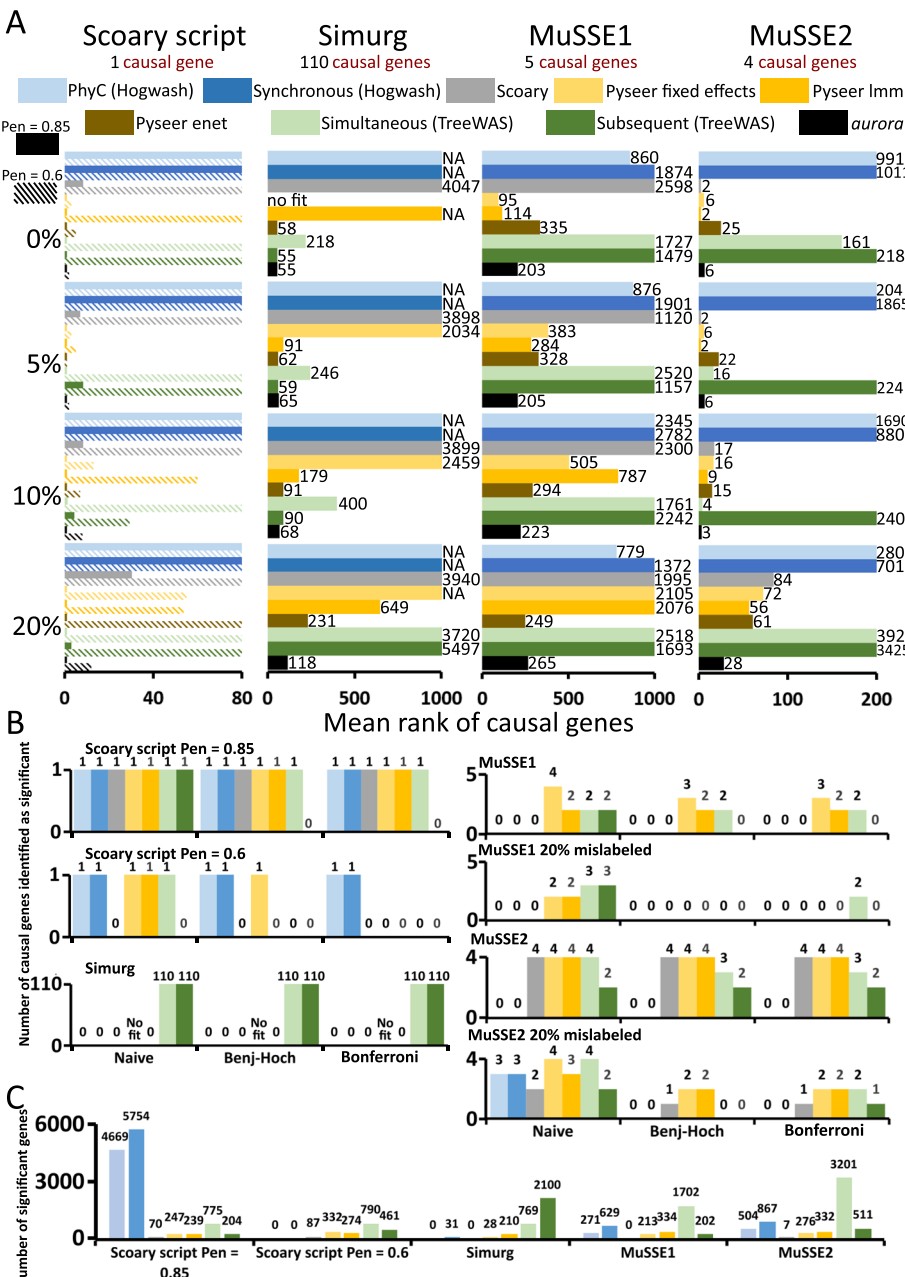

**Fig. 3** Results of analysis of simulated data. Four simulated datasets were constructed using four different methods: Simulate_pan_genome.py a script published along with Scoary (two datasets each with different penetrance 0.85 and 0.6), Simurg, and two multiple state speciation and extinction models—MuSSE1 and MuSSE2. Each simulation was run a single time. **A** Results of the GWAS analysis. In each dataset, 5%, 10%, and 20% of the strains were mislabeled and the GWAS analysis was repeated. The measure of success (*x*-axis) was the mean of ranks of the true causal genes. All genes were first ordered by their *p*-values (or F1 values or standardized residuals) and then the average rank was computed for the causal genes. Four existing mGWAS tools and *aurora* were used for these analyses. **B** The graphs show how many causal genes (*y*-axis) of the simulated trait were identified as significant by the mGWAS tools. There are three groups of results: naïve (no multiple comparison adjustment) and two sets where the *p*-values were adjusted by either the Benjamini–Hochberg method or Bonferroni correction. **C** In this analysis, the phenotype labels were randomized, and the graphs show the number of significant genes (false positives) that each test produced. The graph shows the result of naïve analysis

decreased only moderately from mean rank 55 to 118. Next, we analyzed the performance of all the tools on the MuSSE1 dataset. When no strain labels were swapped, fixed effects and linear mixed model outperformed *aurora* (mean rank values of 95 and 114 respectively vs 203, Fig. 3A). The better mean rank values of the two models are likely a result of their shortcomings. The two models ignore genes that are uniquely present in one lineage, but *aurora* takes these genes into consideration. The MuSSE1 causal gene distribution was chosen so that no causal variant was a present in only one lineage while still being closely tied to the phylogeny. *aurora* thus considers a larger number of variants than the two linear models implemented in Pyseer. As the number of incorrectly labeled strains increased, *aurora* reported consistent scores (203 to 265) while the mean ranks of fixed effects and linear mixed model increased substantially when only 10% of strains had swapped labels (505 and 787 respectively, Fig. 3A). The mean rank value of the elastic net model was stable. Although MuSSE1 and MuSSE2 scenarios are very different, the MuSSE2 dataset showed similar results. Again, we observed stable mean rank values for *aurora* and elastic net (however, the former was lower) and initial good performance of fixed effects and linear mixed model, which, however, worsens with a higher number of incorrectly labeled strains (Fig. 3A).

Taken together these findings demonstrate that *aurora* outperforms other tools in the identification of causal genetic variants especially in cases where a large number of strains have incorrect class labels. These results also confirmed that Hogwash, Scoary, fixed effects, and linear mixed effects models are not able to identify causal genes that are lineage effects. Moreover, the simulated data showed that *aurora* can find and remove the mislabeled strains, and the true causal signal is not lost upon strain mislabeling.

The mean rank value is not sufficient to fully describe the performance of a tool. Previous studies focused on using statistical power as the measure of success [15, 22, 35]. In the context of mGWAS, statistical power describes the probability that a statistical test with a given sample will correctly detect a true causal variant. We thus decided to test if the available tools could identify the causal variants in our simulated data as significant. To adjust the results for multiple comparisons, we applied the most commonly used Benjamini–Hochberg method [36] and the recommended Bonferroni correction [12]. In the latter method, the chosen significance threshold ($\alpha = 0.05$) was divided by the number of non-correlated variants (number of variants after *aurora* prefiltering stage), rather than the total number of variants, to account for the reduced effective number of independent tests. The results are shown in Fig. 3B; *aurora* and elastic net are not included because these methods do not produce *p*-values. With the exception of the Subsequent test, all tools showed the one causal variant in the Scoary script dataset with high penetrance as significant. The Scoary script dataset with medium penetrance was more challenging. Albeit with low rank, Hogwash was the only tool that identified the causal variant as significant in all cases. Except for tests in TreeWAS, no other tools identified all causal genes in the Simurg dataset as significant. Fixed effects model and linear mixed effects model could detect some causal variants in the MuSSE1 and MuSSE2 datasets but when the same analysis was replicated on a dataset with 20% of strains with incorrect class labels most causal variants were not detected as significant. In conclusion, as the results in Fig. 3B vary significantly the ability to detect causal variants relies on the chosen tool. Moreover, output from some of the tools contained a large number

of non-causal significant hits (*p*-value < 0.05). This phenomenon was investigated fur-
ther by assigning the strains of the five above-described simulated datasets into two ran-
domly selected phenotype classes. It is not possible to calculate the expected number of
false positives since it has been shown that the distribution of *p*-values obtained from
mGWAS tests is not uniform [14, 23]. Figure 3 shows that naïve results (no multiple
test adjustment) lead to multiple false positives. The false positives largely diminish after
applying either of the multiple comparison correction methods but both Hogwash tests
and simultaneous test still produce many false positives (Additional file 1: Fig. S3). It
should be noted that *aurora* does not have a significance threshold and thus it cannot
produce any false positives but in all cases, function aurora_pheno() correctly identified
that the strains lack any genomic adaptation towards the randomized phenotype.

### *aurora* is the only GWAS tool that can identify genes responsible for habitat adaptation in *Mycobacterium avium* subsp. *paratuberculosis*

*Mycobacterium avium* subsp. *paratuberculosis* (MAP) is the etiological agent of Johne's
disease, also called paratuberculosis, in ruminants. Johne's disease is a chronic gastro-
intestinal disease accompanied by weight loss, diarrhea, and progressive weakness. In
addition to cattle, MAP is isolated from sheep and goats [37]. MAP has unique genomic
adaptions to both cattle and sheep [38–41] which makes it a good test case for *aurora*.
Mycobacterial genomes are relatively stable with low mutation and recombination rate
[42]. The MAP dataset is therefore similar to the simulated data created by Simurg.

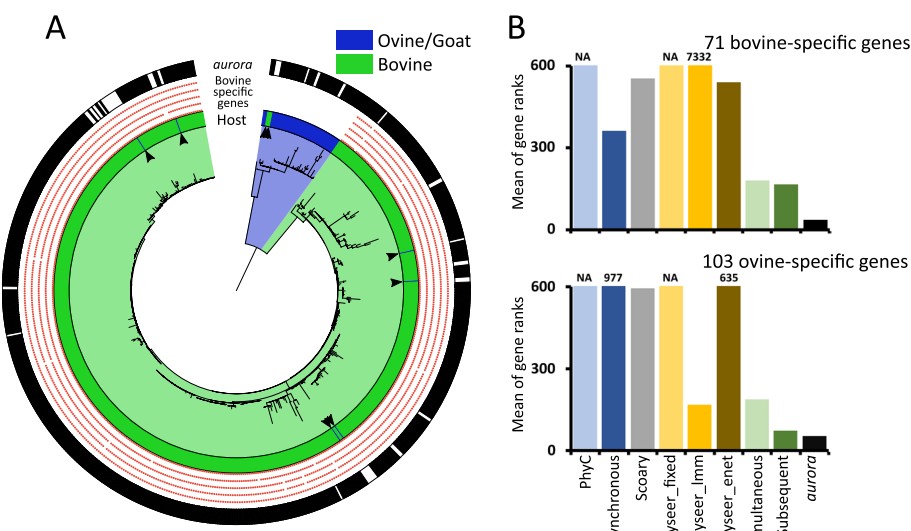

**Fig. 4** GWAS analysis of colonization factors of *Mycobacterium avium* subsp. *paratuberculosis*. **A** Core genome
phylogenetic tree of MAP. The innermost circle shows the host from which the strain was isolated. The next
five circles indicate the presence/absence pattern of cattle-specific essential colonization factors identified
by Eshraghisamani et al. [41]. The outermost circle indicates which strains were identified as autochthonous
(black) and allochthonous (blank) by *aurora*. The arrowheads point to empirically identified allochthonous
strains in the dataset. **B** Result of GWAS analyses of putative bovine and ovine colonization factors. The *y*-axis
represents the mean of gene ranks. All genes were first ordered by their *p*-values (or F1 values in the case of
*aurora*) and then the average rank was computed for the putative colonization genes. The NA means that all
putative colonization factors were not in the output or that they all had a *p*-value equal to 1

MAP strains are divided into two phylogenetic lineages (Fig. 4A), one dominated by cattle isolates (bovine lineage) and one dominated by sheep isolates (ovine lineage). We hypothesized that the ovine isolates present in the bovine lineage are either falsely labeled or are bovine strains ingested by sheep/goats (allochthonous strains). Likewise, the same could occur in the case of the two cattle isolates in the ovine lineage. MAP is prevalent and abundant in cattle and contamination of water or soil is common [37]. Moreover, because the beginning of MAP infection can be subclinical, and the course of the disease is chronic, fecal contamination facilitating transmission of MAP to different hosts is likely to occur. Cattle are relatively resistant to infection by the 'S' (or sheep) type of MAP which causes most cases of ovine and caprine paratuberculosis [43]. This suggests that sheep strains cannot stably colonize cattle; thus, the two cattle isolates in the ovine lineage should be allochthonous. Genotyping analysis showed that the bovine "C" type (or cow) MAP is present in both sheep and goats [44]. C-type is however rare in sheep compared to the S-type [44]. This further points to the possibility of mixed host labels in the dataset. If this hypothesis is true, then we could use the MAP dataset to test the ability of *aurora* to detect mislabeled strains.

To test this hypothesis, we investigated if the cattle isolate genomes in the ovine lineage harbored previously identified essential cattle colonization factors. Likewise, we also investigated if the ovine/goat isolates in the cattle lineage have cattle colonization factors. Eshraghisamani et al. identified a set of 690 genes essential for MAP colonization of cattle [41]. Only 22 out of the 690 genes could not be mapped to the MAP pangenome constructed herein. Out of those that could be mapped to MAP pangenome, 661 colonization factors belonged to the core genome and 2 genes were not more frequent in bovine or ovine isolates. The 5 remaining genes were much more predominant in bovine isolates. The presence of these genes was mapped to the MAP phylogenetic tree (Fig. 4A). These genes are present only in the bovine lineage and absent in the ovine lineage. This confirmed that the two cattle isolates in the ovine lineage are not adapted to cattle since they lack the 5 essential cattle colonization factors. All six ovine/goat isolates present in the bovine lineage had the 5 cattle-specific colonization genes and are thus likely to stably colonize the bovine host.

With the correct labels established, we sought to identify additional genes that are responsible for host adaptation. To this end, genes present in at least 90% of all strains in the bovine lineage and absent in all strains in the ovine lineage were extracted and annotated. Likewise, all genes present in at least 90% of all strains in the ovine lineage and absent in the bovine lineage were annotated (Additional file 3). In total, 71 and 103 putative colonization factors were identified as cattle or ovine-specific respectively. Then, Pyseer, Scoary, TreeWAS, Hogwash, and *aurora* were applied to the MAP dataset. The allochthonous strains identified above were not removed. As in the analysis of simulated data, the mean of the ranks of the colonization factors was taken as the measure of the tool's success rather than the *p*-value produced by the tool. Even though the effect sizes of the bovine and ovine colonization factors are high, most tools were not able to identify the cattle and ovine colonization factors as the highest ranking (Fig. 4B). In contrast, the top results from *aurora* were dominated by these genes. In the first phase, *aurora* correctly identified all the mislabeled strains that we independently identified as incorrectly labeled above using experimental data from Eshraghisamani et al. [41] (Fig. 4A).

Specifically, the Strict mode removed 33 strains which left 492 strains for the subsequent GWAS analysis. All ML algorithms correctly identified MAP as host-adapted. The results from *aurora* sorted by F1 values show that all 71 cattle colonization genes were the top hits, and the 103 ovine colonization factors were all in the top 106 hits. In summary, analysis of MAP showed that *aurora* can be successfully applied to a species with low mutation and recombination rates and to identify causal lineage effects. We were also able to independently identify strains with incorrect class labels in the MAP dataset using experimental results and we demonstrated that *aurora* was able to find and remove those strains (Fig. 4A). As per their performance on simulated data, Hogwash, Scoary, and Pyseer were not able to analyze trait with causal lineage effects (Fig. 4B).

### *aurora* was the only GWAS tool tested that could identify genes responsible for host adaptation in *Salmonella enterica* serovar Typhimurium

*Salmonella enterica* serovar Typhimurium (S. Typhimurium) is a Gram-negative bacterium that is responsible for a significant proportion of foodborne illnesses worldwide. *Salmonella* is a genus with a highly variable genome and some serovars exhibit narrow host specificity while others can infect a broad spectrum of hosts [29, 45]. Serovar Typhimurium has been isolated from multiple hosts such as pigs, humans, cattle, ducks, poultry, and wild avian species [29]. Host adaptation of *S.* Typhimurium is difficult to analyze with the existing mGWAS tools or ML algorithms [46, 47] because some lineages have a wide host range, while others are restricted to one host [29–31] and because strains isolated from humans originated mostly in animals (zoonoses) [47, 48]. Thus, a potentially important colonization factor might be present in only a fraction of the isolates from a particular host. The strains and lineages with a wide host range must therefore be removed or analyzed separately. *S.* Typhimurium is a widespread pathogen and a large number of sequenced genomes are available. This made it possible for us to divide the assemblies into two groups, the first being the discovery group (1223 isolates), which consisted of only high-quality genome assemblies. This group was used to identify colonization factors. The second group was the validation group (4816 isolates) used for validating results from the discovery group.

We first used experimental data to benchmark *aurora* and the existing mGWAS tools. Chaudhuri et al. identified genes essential for the colonization of calves, chickens, and pigs using random insertion mutagenesis and subsequent screening of the mutant pool of strain SL1344 in the host feces [49]. While *S.* Typhimurium SL1344 was originally isolated from cattle, it has also been shown to colonize chickens [50], pigs [51, 52], and even mice [53]. *S.* Typhimurium SL1344 is thus considered a model broad-host strain. However, the results from *aurora* show that strain *S.* Typhimurium SL1344 is adapted only to the bovine host. Interestingly, out of the 50 poultry colonization factors identified by Chaudhuri et al. [49], 47 were present in the *S.* Typhimurium pangenome constructed here and all of these belong to the core genome. Similarly, out of the 63 porcine colonization factors identified by Chaudhuri et al. [49] all could be mapped to a pangenome and 60 belong to the core genome. The remaining three genes were not more predominant in porcine isolates than in isolates from other hosts. On the other hand, out of the 283 cattle colonization genes, 270 could be mapped to the pangenome and 11 of those were in the accessory genome. These 11 genes thus might be the true host-specific colonization

factors. We tested whether *aurora* and other mGWAS tools could identify the 11 bovine colonization factors discovered by Chaudhuri et al. [49]. None of the mGWAS tools (Scoary, Pyseer, Hogwash, and TreeWAS) were able to identify these genes as significant cattle colonization factors. *aurora* on the other hand ranked 8 of these 11 colonization factors in the top 84 genes (ranking based on F1 values). The best gene had a rank of 10.5. In summary, while *S.* Typhimurium SL1344 certainly possesses genes that are essential for mouse, poultry, and porcine colonization [49, 53], these genes are present in all *S.* Typhimurium isolates and not unique to isolates from these hosts. Analysis of SNPs or *k*-mers would be necessary to identify all host colonization factors including those in the core genome. The exceptions are the bovine colonization factors of *S.* Typhimurium SL1344 which are in the accessory genome and which only *aurora* was able to identify. It is possible that the survivability and colonization potential of *S.* Typhimurium SL1344 is higher in the bovine host which is evidenced by the number of colonization factors identified by Chaudhuri et al. and also by the reported higher average read coverage (which however depends on the read coverage in both in input and output pools) [49].

The results from *aurora* analysis of *S.* Typhimurium show that while *S.* Typhimurium is host-adapted, not all strains from the same host share the same adaptation strategy (Fig. 5AB). We utilized the clustered heatmap calculated from random forest proximities (output object of function aurora_pheno()) to identify strains with differing sets of colonization factors (Fig. 5A). The resulting clusters were mapped to a phylogenetic tree (Fig. 5B). Two poultry clusters were identified and named poultry1 and poultry2. Likewise, two bovine clusters (bovine1 and bovine2) with distinct adaptation strategies were found and one porcine cluster (porcine1) was identified. These five clusters were used as new phenotype classes in the function aurora_GWAS() to identify cluster-associated genes.

Next, the validation dataset was used to verify the results from *aurora* and all the other mGWAS tools. The top 2 genes from GWAS analysis of adaptation to each host were selected (total 6 genes) and these genes were used to fit a multinomial log-linear model [54]. The residual deviance of the model was plotted as shown in Fig. 5C. The lower the residual deviance, the better the genes explain the variance of the *S.* Typhimurium validation set. This process was repeated 10 times. With each iteration, 6 new genes were added thus lowering the residual deviance. In the previous analysis (Fig. 5AB), *aurora* detected 5 different clusters belonging to three hosts. The genes were selected from these clusters, but the number of genes was kept the same as for other mGWAS tools. Figure 5C shows that colonization factors selected by *aurora* consistently rank the best (e.g., account for the most variability) out of all the tested mGWAS tools. This shows that the 5 clusters that *aurora* identified using the discovery dataset are present in the *S.* Typhimurium population and that the genes *aurora* identified best describe the host adaptation of *S.* Typhimurium. Hence, we demonstrate that *aurora* is not only a mGWAS algorithm, but it can facilitate deeper understanding of the analyzed trait. *aurora* analysis also identified some hitherto not investigated candidate colonization factors. Annotated list of the top 100 genes associated with each cluster are shown in Additional file 4. This list contains genes with known link to the colonization ability of *S.* Typhimurium and also novel putative colonization factors. The significance of these genes for the colonization potential of *S.* Typhimurium is discussed in Additional file 1.

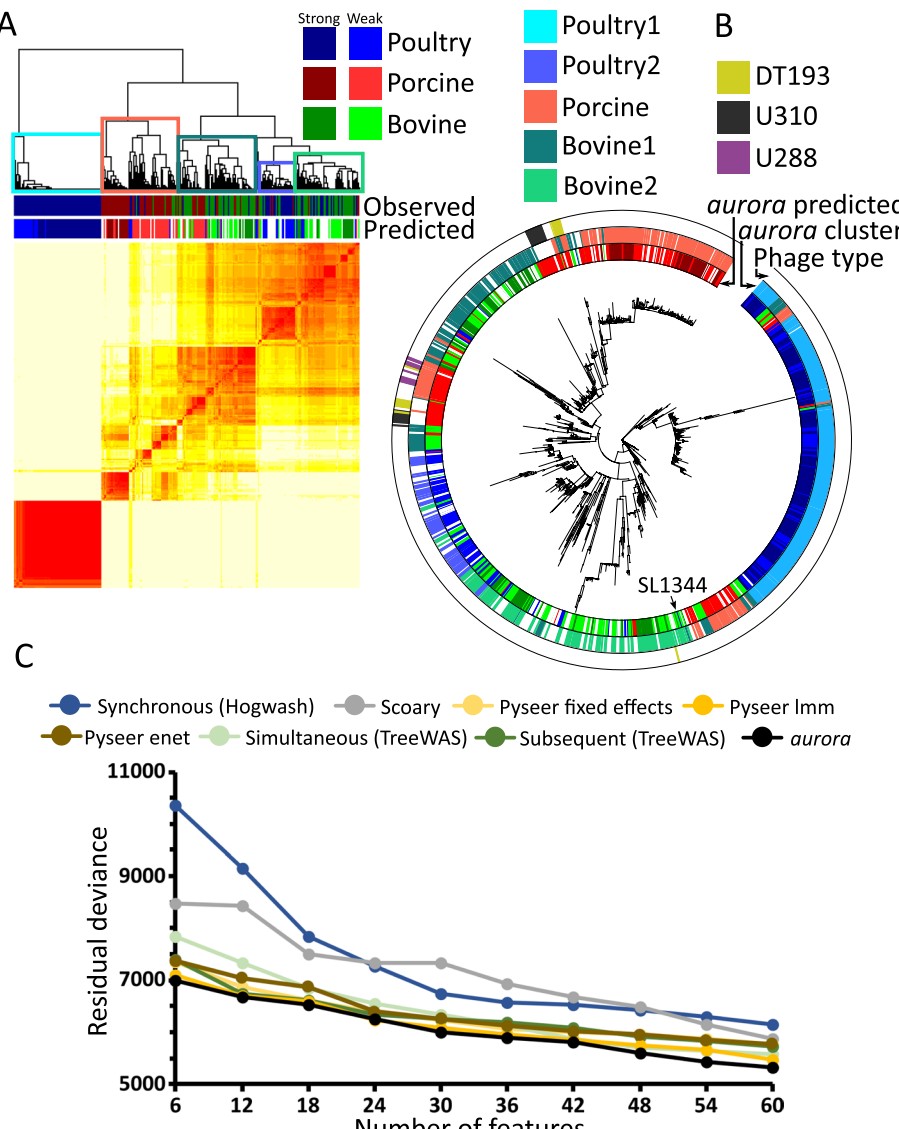

**Fig. 5** GWAS analysis of *Salmonella enterica* serovar Typhimurium. **A** Distance matrix calculated based on random forest proximities obtained in aurora_pheno(). The distance matrix was clustered with hierarchical clustering. The two colored strips show the observed and predicted hosts. If the strain was missing some colonization factors or had some that are typical for other hosts, then it is labeled as a weakly autochthonous (non-typical) strain with a lighter color. If the space is blank, then the strain was found to be allochthonous. Five distinct clusters were identified and mapped to a phylogenetic tree. **B** Core genome phylogenetic tree of *S*. Typhimurium. The inner color strip shows the predicted host. The middle color strip maps the clusters in **A** to the phylogenetic tree. The outer color strip shows the phage types DT193, U310, and U288. **C** Validation of results from *aurora* and other mGWAS tools using the validation *S*. Typhimurium dataset. The *y*-axis shows the residual deviance of a multinomial log-linear model fitted using the top *n* genes (the number of genes is shown in the *x*-axis). Lower residual deviance means that the selected genes better explain the variation in the validation dataset

Another way to verify the results of *aurora* is to examine how *aurora* classified certain phage types. Phage typing of *S.* Typhimurium isolates has been used for decades for surveillance [55]. It is known that some phage types display a broad host range while others are restricted to only one host [29]. Only a few strains in the discovery

dataset were phage typed and out of those, only three phage types (DT193, U310, U288) had a sufficient number of observations. DT193 is a broad-host phage type while U288 is adapted to porcine [29, 56]. Phage type U310 is commonly isolated from porcine, bovine, and wild avian species [57, 58], and it is thus considered a broad-host range phage type. *aurora* correctly predicted that DT193 and U310 strains are not strictly adapted to any habitat since the majority of them were removed prior to the GWAS analysis (Fig. 5B). On the other hand, out of the 15 porcine-adapted U288 strains only two were classified as mislabeled by *aurora*.

### *aurora* can be applied successfully to a variety of phenotypes

As demonstrated above on simulated data, *aurora* performed equally well on phenotypes that were the result of long-term speciation and those that were a result of horizontal gene transfer (Fig. 3A). We wanted to confirm this by analyzing real datasets with phenotypes other than host adaptation. First, two *Neisseria meningitidis* datasets were obtained. One was used for the discovery of genes conferring penicillin resistance and the other for factors of invasiveness. These two datasets had been used in a previous study to validate the ability of package TreeWAS to retrieve causal variants [23]. As reported for TreeWAS in that study, *aurora* was not able to identify any penicillin resistance genes using a pangenome matrix. However, when core genome SNPs were analyzed, the output from *aurora* showed that the top 49 SNPs with the highest standardized residuals mapped to either *murE* or *penA* (Additional file 1: Fig. S4). Both genes were previously identified as key for penicillin resistance [59]. When the invasiveness of *N. meningitidis* was previously analyzed using TreeWAS with reference to then available literature by Collins and Didelot (2017), three genes were identified as invasion factors: *nadA*, *mafA2* (Neisserial adhesins), and *hmbR* (Hemoglobin receptor protein). *aurora* successfully identified *nadA* (rank 34), but it was not able to identify *hmbR* and *mafA2* because these belong to the core genome in our dataset. However, *aurora* identified other genes (Additional file 5). The top invasiveness factor was PorB (outer membrane protein) whose role in host–pathogen interaction is known [60]. *aurora* also identified gene *tbpB* (rank 5) whose product—transferrin-binding protein B—was identified previously as a meningeal host-colonization factor [61].

Next, we tested if *aurora* could retrieve the genetic determinants of extra-intestinal virulence of the species *Escherichia* previously identified by Galardini et al. using Pyseer and Scoary [26]. That study reported that the high-pathogenicity island (HPI) encoding the yersiniabactin siderophore, aerobactin (iron chelator) operon, and the *sitABCD* operon encoding a $Mn^{2+}/Fe^{2+}$ transport system were the strongest determinants of extra-intestinal virulence. *aurora* was run with the same dataset and the top results also contain these genes. Additionally, *aurora* identified more genes that could be important for extra-intestinal virulence and that the previous study did not discuss. For example, the top results include a permease AmpG whose removal increases beta-lactam susceptibility in *Escherichia coli* [62]. *aurora* also identified a gene that seems to be markedly absent in extra-intestinal virulent *Escherichia* isolates. This gene encodes Cytochrome $b_{561}$ a protein that was previously speculated to be involved in bacterial competition [63]. This is understandable, considering that the colonization of extraintestinal sites does not involve direct bacterial competition and loss of this gene would thus not impair

extraintestinal colonization. The top genes identified by *aurora* are annotated in Additional file 6. Taken together, we utilized simulated and real datasets to demonstrate that *aurora* can successfully analyze a variety of different phenotypes, and therefore the user does not have to make assumptions about the effect of the trait on the phylogeny and about the distribution of the causal variants.

### *aurora* was able to correctly identify host-adapted and mislabeled strains of *Limosilactobacillus reuteri* and *Lactiplantibacillus plantarum*

*Limosilactobacillus reuteri* (formerly *Lactobacillus reuteri*) is a Gram-positive, rod-shaped bacterium that inhabits the gastrointestinal tract of mammals. Because of its ability to colonize and persist in the gut of experimental animal models, *L. reuteri* has served as a model species for studying microbial adaptations to the mammalian gut [64]. Intensive studies have investigated the phylogenetic structure of the species, establishing host-specific phylogenetic lineages, and the gene content within these lineages [65, 66]. The importance of several genes that allow *L. reuteri* to colonize rodents have been experimentally validated [16, 64, 66–72]. These genes were used for benchmarking of *aurora* and other mGWAS tools (below).

First however we sought to identify the mislabeled strains in the *L. reuteri* dataset. To this end, all available high-quality *L. reuteri* genomes were obtained. Out of the 207 genome sequences collected, 90 were rodent isolates. The dataset also contained human (27 strains), poultry (39 strains), porcine (26 strains), and primate (25 strains) isolates. When aurora_pheno() was run, all of its in-built ML algorithms agreed that *L. reuteri* is a host-adapted species (Additional file 1: Fig. S1). This was further supported by defined clusters in the clustered heatmap derived from random forest models (Fig. 6A) and from CART models (Additional file 1: Fig. S5B). The random forest models supported division of strains into clusters for each of the 5 hosts (Fig. 6A). Since the *L. reuteri* rodent colonization factors are known, we examined the *aurora* results and verified that the predicted labels align with the expected outcomes. This is discussed in detail in Additional file 1, and more information on host adaptation genes appears below. Additionally, we were also interested if the ML algorithms agree on which strains are allochthonous (Fig. 6B). With the exception of log regression which suggested removing more strains (Fig. 6C), there was a general consensus on which strains are allochthonous, with some degree of variation. As discussed in Additional file 1, the ML algorithms complement each other and together they are able to capture all allochthonous strains.

*Lactiplantibacillus plantarum* (formerly known as *Lactobacillus plantarum*) is a gram-positive, rod-shaped, and non-spore-forming bacterium that is commonly detected in various ecological niches, including plant materials, fermented foods, dairy products, and the gastrointestinal tracts of animals and humans [73]. In contrast to *L. reuteri*, *L. plantarum* is considered a generalist species without any specialization to a particular habitat [74, 75]. However, some studies have suggested that there may be some level of habitat specific adaptation. For example, *L. plantarum* strains isolated from the human gut and pickled cabbage have been shown to have specific gene structural variations, which may help them to survive and compete in these two different environments [76]. On the other hand, clustering based on phenotypic properties showed that human isolates are scattered in multiple clusters which points to the fact that human isolates

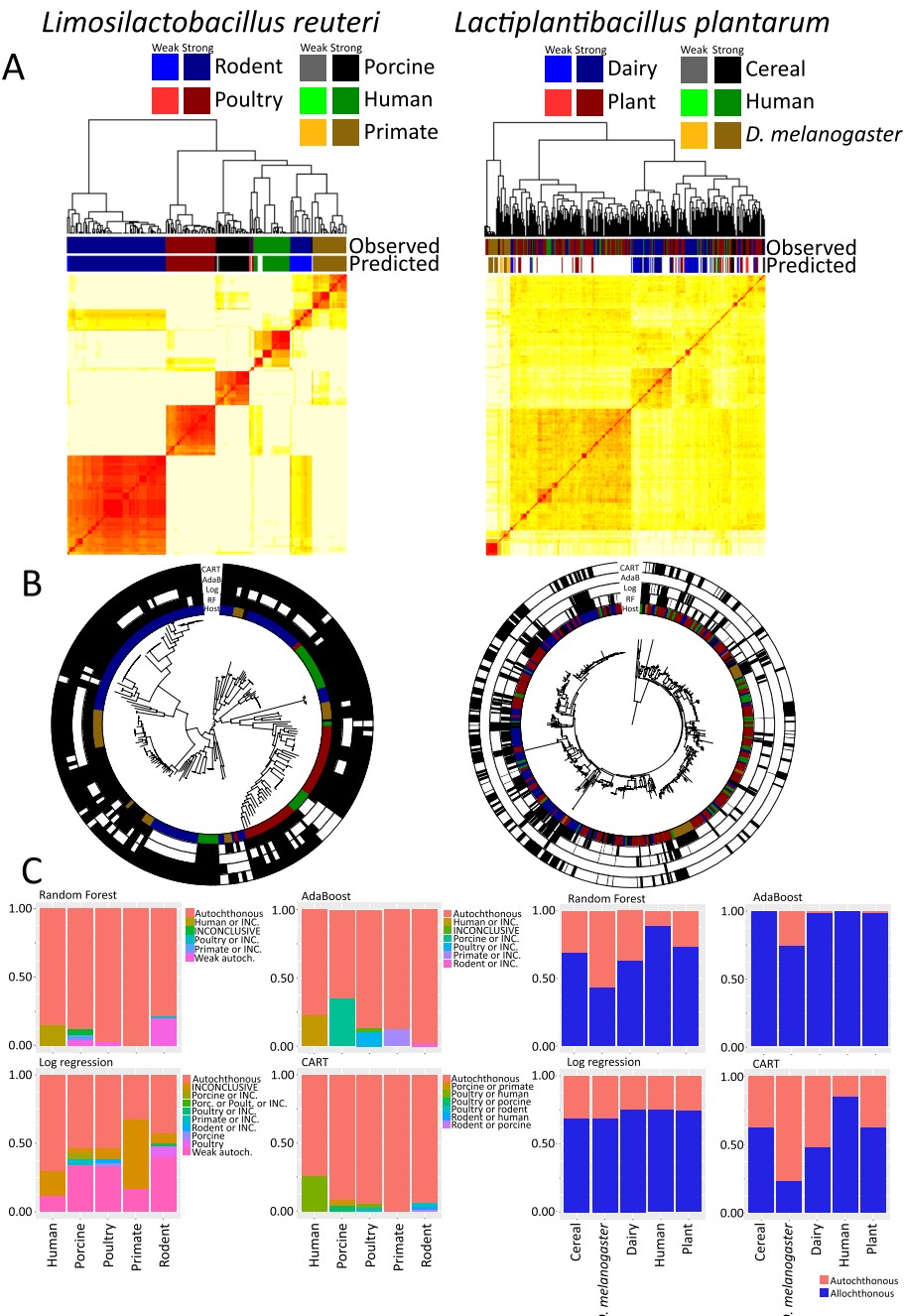

**Fig. 6** Results of *aurora* applied to *L. reuteri* and *L. plantarum* datasets. **A** Distance matrix calculated based on random forest proximities obtained in aurora_pheno(). The distance matrix was clustered with hierarchical clustering. The two color strips show the observed and predicted hosts. If the strain was missing some colonization factors or had some that are typical for other hosts, then it is labeled as a weakly autochthonous (non-typical) strain with a lighter color. If the space is blank, then the strain was found to be allochthonous. **B** Core genome phylogenetic tree of *L. reuteri* and *L. plantarum*. The innermost strip shows the habitat from which the strain was isolated. The four outer strips show which strains were identified as allochthonous (mislabeled) by *aurora* (blank) and which strains were identified as autochthonous in the observed habitat (black). **C** The graphs show detailed predicted labels and the portion of the strains associated with each label

are not specifically adapted to the human GIT [77]. A small-scale comparative genomics study showed that the human isolate *L. plantarum* ZJ316 has multiple enzymes and pathways that are absent in strains from other sources pointing to a specific adaptation towards mammals [78]. Another small-scale comparative study identified differences between *L. plantarum* isolates from human GIT, dairy, and food [79].

Most of the studies that investigate habitat adaptation of *L. plantarum* did not control for population structure which is a major confounder in microbial comparative studies [2]. Furthermore, most studies analyzed only a small number of strains. This leads to underpowered analyses and results that cannot be generalized to the whole *L. plantarum* population. Since *L. plantarum* may be isolated from multiple overlapping ecological habitats, it is clear that strains may shift between habitats. Indeed, multiple studies suggest that *L. plantarum* strains isolated from the human GIT are not specifically adapted to this environment [75, 77, 80]. Additionally, a study of hundreds of human, environmental, and food metagenome-assembled genomes (MAGs) showed that *L. plantarum* is neither prevalent nor abundant in the GIT of humans and it is likely not a long-term resident there [81, 82]. Probiotic trials also showed that probiotic *L. plantarum* strains colonize the human gut only transiently [83–85]. Because food is a common source of *L. plantarum*, it is likely that the human gut isolates were ingested with the food.

*aurora* was applied to the *L. plantarum* dataset to test if it can identify human *L. plantarum* isolates as an allochthonous population. Five hundred fifty five high-quality genomic assemblies of *L. plantarum* were downloaded and annotated. Out of those, 60 were human isolates. The dataset was dominated by plant isolates—255 (only edible plants were considered) and dairy isolates (173). Additionally, the *L. plantarum* dataset also contained 32 strains isolated from cereals and 35 from *Drosophila melanogaster*. A pangenome matrix and a core genome phylogenetic tree were used as input into *aurora*. The clustered heatmap constructed using proximities derived from random forest models did not reveal defined clusters as had been identified for *L. reuteri* (Fig. 6A). Specifically, human isolates were spread across the clustered heatmap. The *p*-value matrix from random forest models showed that human strains are indistinguishable from plant and cereal isolates (Additional file 1: Fig. S2). Random forest models also indicate that only seven out of 60 human isolates are autochthonous in this habitat. On the other hand, dairy and *D. melanogaster* isolates are significantly different from each other and from the other habitats. The *D. melanogaster* isolates are the only isolates that seem to form one almost continuous cluster (Fig. 6A). Additionally, CART models predict that 32 out of the 35 *D. melanogaster* isolates are autochthonous in this environment. However, it should be noted that the *D. melanogaster* strains in the dataset belong to only two lineages with short branches (Fig. 6B). It was shown that the adaptation of *L. plantarum* is influenced by the diet of *D. melanogaster* rather than by the host itself [86]. The formation of these two lineages could be a result of a differential diet of two *D. melanogaster* populations. To establish if *L. plantarum* adapts to *D. melanogaster*, a larger and more diverse dataset would have to be gathered. The size and diversity of human isolates are, on the other hand, sufficient for this analysis. Random forest thus confirmed that *L. plantarum* strains isolated from human GIT are not adapted to this environment.

Log regression analysis generated a similar finding. According to this algorithm, with the exception of the *D. melanogaster* host, none of the analyzed classes can be

distinguished (Additional file 1: Fig. S2). Log regression also confirmed that human strains are not adapted to their isolation habitat. Only 15 strains out of 60 were classified as autochthonous in the human GIT (Fig. 6C). The results from AdaBoost and CART are consistent with the previous two algorithms. AdaBoost did not identify even one human autochthonous isolate while CART identified 18 such strains.

Figure 6B and C show that if all ML algorithms were considered (e.g., a Strict mode in aurora_GWAS()) almost all *L. plantarum* strains would be classified as mislabeled. Only two strains were classified as autochthonous by all algorithms. Thus, if the user wishes to run GWAS analysis on this dataset, then a Consensus mode would have to be applied. In this mode, only strains that are identified as mislabeled by all applied ML algorithms are removed. This mode removed a total of 237 strains and there are 318 strains left for the GWAS analysis. Despite the lack of adaptation to most habitats, we annotated the top 50 genes for each habitat (Additional file 7). Taken together, *aurora* demonstrated that it can correctly recognize that most *L. plantarum* strains do not have strict habitat range.

### Benchmarking mGWAS on *L. reuteri* and *L. plantarum* datasets

Given that the adaptation mechanisms of *L. reuteri* to the rodent GIT are known, we aimed to evaluate *aurora's* ability to find colonization factors compared to other mGWAS tools. Urea is abundant in the rodent gut, and the urease operon has been shown to contribute to ecological performance by increasing acid resistance of *L. reuteri* in the stomach, which is its primary niche in rodents [87]. Another critical adaptation of *L. reuteri* to the gut environment is the production of biofilms. *L. reuteri* forms biofilms on the gastric epithelium of the forestomach, allowing it to persist and compete with other gut microbiota members. *L. reuteri* produces an extracellular matrix [88] and surface proteins that allow the cells to aggregate [70]. Frese et al. showed that a surface adhesin—SRRP (serine-rich repeat protein) whose export is enabled by the SecA2-SecY2 pathway allows initial adhesion of cells in the rodent gut and is essential for biofilm formation, colonization, and competition [89]. These are thus essential genes for rodent colonization. Additionally, the presence of glutaminase genes can further enhance acid resistance in the rodent isolate *L. reuteri* 100–23 [72].

Next, we wanted to investigate whether these genes were exclusively present in rodent isolates. The presence of the urease (*ure*) operon in the pangenome was very selective and specific for rodent isolates. Eighty nine out of 90 rodent genomes harbored a complete *ure* operon. One strain did not possess any of the *ure* genes except for the urease accessory protein *ureD*. The complete *ure* operon was also present in 2 (5% of all poultry strains) poultry isolates, 8 (30%) human isolates, and 10 (40%) primate isolates. The SecA2-SecY2 dependent SRRP was present in all 90 rodent isolates. Additionally, the adhesin was present in 3 (8%) poultry, 7 (26%) human, 2 (8%) porcine, and 12 (48%) primate isolates. The SecA2-SecY2 protein translocation pathway genes were not as specific to rodent strains. The majority of rodent isolates (88 out of 90) had the secretion pathway genes as well as 3 (8%) poultry, 6 (22%) human, 14 (56%) primate strains, and 22 out of 26 porcine isolates (85%). There were two glutaminase genes present in the pangenome. The first, *glsA*, was present in almost all isolates across the hosts, while the second, *glsA_1*, was present in 47 out of 90 rodent isolates and mostly absent in isolates from other hosts. The latter gene *glsA_1* and

the SecA2-SecY2 pathway are thus low effect size variants without full penetrance and are thus valuable test examples for the sensitivity of mGWAS tools. In summary, a successful mGWAS tool should be able to identify the *ure* operon, genes for the SecA2-SecY2 pathway, the genes encoding SRRP, and glutaminase *glsA_1* as significant rodent gut colonization factors of *L. reuteri*. These four colonization factors were used for benchmarking of mGWAS tools.

Pyseer, Scoary, TreeWAS, Hogwash, and *aurora* were run with both *L. reuteri* and *L. plantarum* datasets. Some of these tools do not implement multiclass problems, and thus, only the adaptation of *L. reuteri* to rodent GIT and *L. plantarum* to human GIT was analyzed. Scoary produces two *p*-values per gene: a *p*-value from Fisher's exact test that is adjusted for multiple comparisons and a population-aware empirical *p*-value. Only genes for which both *p*-values were below 0.05 were considered significant. Scoary identified 425 significant rodent colonization factors. However, the *ure* operon, SecA2-SecY2 pathway, and *glsA_1* genes were not among the significant genes. While Fisher's exact test *p*-values of these genes were below 0.05, the empirical *p*-values were between 0.97 and 0.2. Only the SRRP was correctly identified as significant with an empirical *p*-value of 0.02.

The fixed effects model used by Pyseer on the other hand correctly identified the *ure* operon, SecA2-SecY2 pathway, and SRRP as significant. The genes in the *ure* operon were among the most significant genes, and similarly, the SRRP had the fourth lowest *p*-value (Table 1). However, these genes are not difficult to identify since their effect sizes are large. The fixed effects model failed to identify the low effect size *glsA_1* which had an unadjusted *p*-value $= 0.26$. The *p*-value of the SecA2-SecY2 pathway passed both the Bonferroni correction and Benjamini–Hochberg method; however, the two genes ranked very low (133.5, Table 1). The linear mixed effects model failed to identify all genes as significant. The lowest unadjusted *p*-value was that of the *ure* operon ($< 0.01$) but this was not enough to pass Benjamini–Hochberg or Bonferroni adjustments. The unadjusted *p*-values of all the other genes were between 0.4 and 0.96. The elastic net model had non-zero slopes for all the genes and the ranks of the *ure* operon, SecA2-SecY2 pathway, and the SRRP were between 10.5 and 43. On the other hand, the rank of *glsA_1* was only 143.5 (Table 1). It should also be noted that Pyseer labeled the *ure* operon genes and the SRRP with "bad-chisq" label. This means that (i) these genes have large effect sizes and (ii) the initial prefiltering was not stringent enough. It is a common practice to discard such genes [35] which further points to the inappropriateness of this method for this dataset. We also computed the lineage effect association test using the *bugwas* method [11]. However, none of the principal components were significantly associated with the rodent host.

The Simultaneous test implemented in the TreeWAS package identified 429 genes with unadjusted *p*-value $< 0.05$ with *ure* operon, SRRP, and *glsA_1* included (Table 1). The *ure* operon genes had one of the lowest *p*-values. However, after the Benjamini–Hochberg procedure or Bonferroni adjustment, none of these genes were significant. Moreover, the SecA2-SecY2 pathway genes had an unadjusted *p*-value $> 0.3$. Using the Subsequent test, only the *ure* operon and SRRP were significant after both the Benjamini–Hochberg method and Bonferroni adjustment. All the remaining genes pass neither significance threshold after adjustment.

**Table 1** Summary of GWAS analyses of *L. reuteri* rodent adaptation genes. The rows show experimentally verified *L. reuteri* adaptation genes and columns all tested mGWAS tools. There are three measures of success that were chosen: Benjamini–Hochberg method, Bonferroni adjustment and rank in the sorted results. The pipe symbol (✓) indicates that the genes were identified as significant

| Colonization factor | Measure | Scoary[a] | Fixed effects | Linear mixed effects | Elastic net[b] | Simultaneous | Subsequent | Synchronous | PhyC | *aurora* |
|---|---|---|---|---|---|---|---|---|---|---|
| *ure operon* | Benjamini-Hochberg | ✓ | ✓ | X | NA | X | ✓ | ✓ | X | NA |
| | Bonferroni | ✓ | ✓ | X | NA | X | ✓ | ✓ | X | NA |
| | Rank | 1324.5 | 14.5 | 34.5 | 10.5 | 28.5 | 10.5 | 29.5 | NaN | 9.5 |
| Serine-Rich Repeat Protein (SRRP) | Benjamini-Hochberg | ✓ | ✓ | X | NA | X | ✓ | X | X | NA |
| | Bonferroni | ✓ | ✓ | X | NA | X | ✓ | X | X | NA |
| | Rank | 285 | 4 | 513 | 17 | 140.5 | 16 | 246 | 3016 | 18 |
| SecA2-SecY2 pathway | Benjamini-Hochberg | ✓ | ✓ | X | NA | X | X | X | X | NA |
| | Bonferroni | ✓ | ✓ | X | NA | X | X | X | X | NA |
| | Rank | 1504.5 | 133.5 | 3086.5 | 41 | 4877 | 52.5 | 2642.5 | NaN | 32.5 |
| Glutaminase (*glsA_1*) | Benjamini-Hochberg | ✓ | X | X | NA | X | X | ✓ | X | NA |
| | Bonferroni | ✓ | X | X | NA | X | X | ✓ | X | NA |
| | Rank | 2143.5 | 1994.5 | 685 | 143.5 | 140.5 | 74.5 | 64 | 301 | 41.5 |

*NA* the tool does not compute *p*-values; *NaN* the gene was not in the results produced by the tool

[a] Genes were sorted based on empirical *p*-value

[b] Genes were sorted based on filter *p*-value

The Synchronous test implemented in Hogwash identified 223 significant genes based on unadjusted *p*-values with *ure* operon and *glsA_1* having one of the lowest *p*-values. Only these two genes were significant even after the Benjamini–Hochberg procedure and Bonferroni adjustment (Table 1). PhyC test identified 220 significant associations. However, neither of the genes discussed herein are among those. PhyC test prioritizes genes that are only present in rodent isolates and absent everywhere else, but this comes at the expense of sensitivity.

In summary, as was the case for our analysis of simulated data, we found major differences between the results from all the tested tools even though all of them have the same hypotheses. The results are summarized in Table 1. Based on the ranks of the known colonization factors, only the Subsequent test implemented in the TreeWAS package would be suitable for the identification of all rodent colonization genes in *L. reuteri*. None of the existing mGWAS tools would be able to identify the four colonization factors after multiple comparison adjustments. Next, *aurora* GWAS analysis was run on an *L. reuteri* dataset where all mislabeled strains were removed. As shown in Table 1, *aurora* was the only tool that assigned a low rank value to the four investigated colonization factors. Overall, *aurora* was the only tool that identified the four colonization factors in the top 50 results (Table 1). We have annotated the foremost 50 colonization factors across all hosts, and these results are available in Additional file 8.

We also benchmarked Pyseer, Scoary, TreeWAS, Hogwash, and *aurora* on the *L. plantarum* dataset. Only adaptation towards the human GIT was evaluated. As discussed above, *L. plantarum* is a generalist species, and fermented food, cereals, and dairy are likely the original sources of human GIT isolates. Thus, specific genomic adaptation to the human gut should not be detected and GWAS analysis should not lead to any significantly associated genes. However, several tools produced a number of significant genes (unadjusted *p*-value $< 0.05$). Scoary outputs 208 such genes. Even after adjusting for multiple tests, Scoary still predicts that 134 and 13 genes (adjusted by the Benjamini–Hochberg method and Bonferroni respectively) are significantly associated with the human host. To test if these associations are purely accidental, we created a new artificial phenotype where two classes were assigned randomly (random phenotype). Even in this case, Scoary still predicted that 34 genes are significant (lowest *p*-value $= 0.005$).

The fixed effects model and linear mixed model performed similarly. In the case of the random phenotype or the human *L. plantarum* isolates, the former did not identify any significant genes after multiple comparisons adjustment. After Bonferroni correction, the linear mixed model identified only two genes significantly associated with the human host. The Subsequent test on the other hand predicted more than 6000 genes (after Benjamini–Hochberg adjustment) as significantly associated with both the random phenotype and the human host. The Simultaneous test predicted that over 500 genes are significantly associated with the two phenotypes; however, only a few remained significant after multiple comparison adjustments. Both tests in the Hogwash package correctly failed to identify any significant genes associated with the random phenotype, but the Synchronous test predicted over 1000 genes significantly associated with human hosts even after multiple comparison adjustments. Lastly, *aurora* correctly predicted that no adaptation exists for both phenotypes. Taken together, as in the case of simulated data (Additional file 1: Fig. S2), this analysis has shown that TreeWAS and

Hogwash produce multiple false positives when the set of strains is not genomically adapted to the analyzed trait. On the other hand, *aurora* and two models in Pyseer correctly predicted the absence of adaptation.

## Discussion

*aurora* is a comprehensive analysis tool implemented in the R programming language (https://cran.r-project.org/) with multiple hyperparameters that can be modified by more experienced users to suit their needs. However, *aurora* either provides or estimates a set of parameters that will accompany most datasets and it is thus easily executed even by users with limited experience with R and machine learning. The design of the *aurora* package allows for combining the results of function aurora_pheno(), which identifies allochthonous strains and strains with incorrect class labels, with other mGWAS tools. *aurora* provides wrapper functions to run TreeWAS [23] and Hogwash [14] and functions that produce input to Pyseer [21] and Scoary [22]. Common workflows using the *aurora* package are demonstrated here (https://dalimilbujdos.github.io/aurora/).

### *p*-values produced by mGWAS tools are not accurate

Our results revealed several problems inherent in the other mGWAS tools under examination (Pyseer, Scoary, TreeWAS, and Hogwash). The accuracy of *p*-values produced by these mGWAS tools varies greatly depending on the dataset. This has been previously noted, where a study focused on *Staphylococcus aureus* showed that linear regression model utilized by a GWAS tool PLINK [90] yielded many false positives and mGWAS-specific test yielded many false negatives [27]. Permutation methods (i.e., Scoary, Tree-WAS, or Hogwash) can also lead to frequent false positives ([91], synonymous SNPs in this case). Similarly, a benchmarking study showed that some GWAS tools when applied to microbial data produce results with a high rate of false positives [35]. Additionally, simulated data showed that the *p*-values produced by the mGWAS tools are not reliable nor consistent with each other (Fig. 3B,C) which is unexpected since the test assumptions in mGWAS are minimal [2] and they all have the same null hypothesis: the genetic variants do not influence the trait. *aurora* does not produce any *p*-values or significance threshold. Instead, it produces two association metrics: F1 values and standardized residuals. The latter are correlated with *p*-values of $\chi^2$ test only when the number of analyzed categories is two. The relationship between standardized residuals and *p*-values in a $\chi^2$ test becomes more complex as the number of categories increases. Throughout the manuscript, we demonstrate that these metrics are a better measure of association than *p*-values (Figs. 3, 4, and 5, and Table 1). The use of two association metrics has its justification. Just like effect size, the standardized residuals quantify the magnitude of an observed effect. However, the magnitude of an effect is not enough to assess association significance. Therefore, it was proposed to combine effect size estimates with frequency cutoffs to filter out non-causal variants [9]. Because the frequency of variants needs to be considered when assessing the strength of genotype–phenotype associations *aurora* additionally calculates F1 values for each variant. F1 values implicitly consider the frequency of a variant (Additional file 1: Fig. S6) and thus no subjectively chosen frequency cutoff is necessary. The focus of the F1 metric on high frequency variants can obscure the identification of low-effect size variants with low frequency which may be ranked

under non-causal variants. We thus envision that the two metrics that *aurora* calculates should be used in two different scenarios. In some mGWAS works, the goal is to identify a key mechanism underlying the observed trait [16, 26, 92]. In these studies, the F1 value should be prioritized. On the other hand, some studies try to investigate all potential genetic contributions to the phenotype [11, 13, 91, 93]. In such a case, ranking variants based on standardized residuals while removing those with low frequency or low recall value is the recommended approach.

An additional benefit of *aurora* also rests in its ability to visualize complex datasets using clustered heatmaps. Visualization is a key step in human GWAS where the analysis relies on Manhattan plots [94]. The applicability of Manhattan plots in microbial mGWAS is constrained due to the lower conservation observed in microbial genomes compared to human genomes. We demonstrate the utility of the clustered heatmaps, which offer a nuanced perspective on trait distribution and the interconnectedness of microbial strains.

### Multiple testing correction methods cannot work with mGWAS data

An additional challenge linked to the issue of accuracy is that currently, there is no consensus on handling the multiple testing problem in mGWAS [2]. Some studies and tools use Bonferroni correction [11, 13, 21, 22] and some use the Benjamini–Hochberg method [10, 22]. However, significant disparities exist in the outcome when these two commonly used methods are applied (Fig. 3C, Additional file 1: Fig. S3). Bonferroni correction is more stringent and often leads to false negatives in the result. Benjamini–Hochberg method may on the contrary lead to frequent false positives. Another problem is that both methods assume feature independence. Due to LD and population structure, it is not possible to fulfill this requirement. When Bonferroni correction is used, it was recommended to group features with identical presence/absence pattern into one and thus reduce the multiple testing burden [12]. This will still lead to a large number of false negatives because feature non-independence persists. If *aurora* users want to use Bonferroni correction to calculate a new significance threshold, which would be applied to results of other tools since *aurora* does not produce *p*-values, we recommend using the number of features after data filtering and grouping done by aurora_pheno(). These initial steps ensure that only unique informative features are preserved. This pre-calculated Bonferroni threshold is part of the aurora_pheno() results.

### Publicly available data contain mislabeled strains

All real datasets used in this study were sourced from public databases, and each was found to include instances of mislabeled strains. It is thus not uncommon to encounter allochthonous strains and assemblies with erroneous metadata. A case in point was illustrated with *Mycobacterium avium* subsp. *paratuberculosis*, where strains despite being well adapted to their hosts were shown to circulate among multiple hosts (Fig. 4 [43, 44]). The primary repositories for information on strain origin or any other phenotype are the BioProject and BioSample databases [95, 96]. However, entries in these databases are manually entered and lack standardization [97]. Consequently, any endeavor to acquire a dataset for mGWAS is susceptible to human errors, either on the part of the submitter or the mGWAS researcher. Even if all metadata are accurate, contamination or

taxonomic misclassification can still pose challenges. Various methods can be employed to exclude affected assemblies before mGWAS analysis [98, 99], but there is a risk that these methods may not be stringent enough. Moreover, sequence and metadata errors are also propagated to secondary databases, making it difficult to trace the sample's origin [97]. With the increasing number of sequenced bacterial genomes, manual verification of metadata correctness for all dataset entries becomes an impractical task. It is anticipated that artificial intelligence will play a role in labeling strains [100]. However, it is important to note that using these models may introduce additional errors. Therefore, the ability of *aurora* to detect and eliminate mislabeled strains (allochthonous strains or strains with incorrect class label) is crucial. Throughout this paper, we demonstrate that removing mislabeled strains prior to mGWAS analysis significantly enhances the ability to identify causal variants.

### *aurora's* GWAS approach operates without the presumption that the trait did not impact the phylogeny

In addition to its ability to discern mislabeled strains, *aurora* also provides an enhanced mGWAS method. One of the most significant breakthroughs in the development of mGWAS tools came with the recognition that microbial traits are influenced by both lineage and locus effects [11]. This realization resulted in the development of a method known as *bugwas*, capable of pinpointing lineage-level associations [11]. Notably, upon analyzing simulated datasets, we showed that most other mGWAS tools are primarily focused on identifying locus effects alone (Fig. 3A). In practice, lineage and locus effects are analyzed separately [13]. Frequently, a variant's occurrence correlates closely with the phylogeny, yet the variant is not uniquely confined to one lineage. As a result, such a variant is not strictly a locus nor lineage effect, and the categorization of variants into these groups is essentially arbitrary. A key motivation behind the creation of *aurora* was to simultaneously analyze lineage and locus effects, facilitating a direct comparison of their impact on the phenotype. This combined with the removal of mislabeled strains prior to GWAS analysis allowed *aurora* to surpass all other mGWAS tools on both simulated and real datasets. Furthermore, we showed that *aurora* can be used for analysis of a variety of phenotypes, and it is thus the most flexible tool that does not assume any causal variant distribution.

### *aurora* was able to identify experimentally verified colonization factors

To validate the results of *aurora* and to benchmark it against other tools, we carried out seven comprehensive mGWAS analyses on real datasets. These datasets have various characteristics. Host adaptation of MAP is facilitated by lineage effects and the dataset includes a few mislabeled strains; the S. Typhimurium dataset exemplifies a case with locus effects and many mislabeled strains; *L. reuteri* shows a mix of lineage and locus effects with many mislabeled strains, and *L. plantarum* is a generalist species. We annotated the top genes within each phenotype class of each dataset, revealing intriguing genotype–phenotype associations. Notably, certain causal variants uncovered were experimentally characterized before, exemplified by the well-established roles of *gogB* and *ssrB* as *S.* Typhimurium colonization factors [101, 102]. The identification of known colonization factors further supports the utility of *aurora*, and discussion in Additional

file 1 thoroughly examines the identified *S.* Typhimurium and *L. reuteri* colonization factors. Lists of annotated variants of each mGWAS dataset are available as Additional files.

## Conclusions

In mGWAS, currently available tools make the erroneous assumptions that all causal variants are present in multiple phylogenetic lineages and that the phenotype was not influenced by the evolutionary history of the species. However, these assumptions are valid only for a specific subset of microbial traits. To address these limitations, we have developed and rigorously tested an R package named *aurora*, designed to handle typical mGWAS confounders and operate independently of assumptions regarding the distributions of causal variants or the phenotype. This makes *aurora* the most versatile mGWAS tool currently available. Through extensive testing with both simulated and empirical datasets, we demonstrate *aurora's* efficacy in identifying causal genetic variants, regardless of whether they manifest as locus or lineage effects. Additionally, we show that publicly available datasets contain allochthonous strains and strains associated with erroneous metadata. *aurora* utilizes machine learning algorithms to identify and remove these strains, thereby enhancing the power to detect genuine genotype–phenotype associations. This functionality is independent of the subsequent mGWAS analysis, allowing users to utilize *aurora* for strain filtering before employing other mGWAS tools. Importantly, *aurora* can identify cases where an entire species does not possess genomic variants associated with the analyzed phenotype, a feature some mGWAS tools fail to recognize, resulting in numerous false positives. Notably, *aurora's* GWAS method does not rely on any determined significance threshold, a crucial consideration in mGWAS due to the high collinearity of analyzed variants and the inherent limitations of methods attempting to disentangle these collinearity patterns. Despite these improvements, we acknowledge that *aurora* has its limitations. When a phenotype is determined solely by lineage effects, as seen in the MAP dataset presented here (Fig. 4), linkage disequilibrium can lead to inflated association metrics of non-causal variants. Furthermore, if there is an imbalance in the number of strains across phenotype classes, *aurora* may oversample the less numerous category, resulting in inflated association metrics. Hence, we strongly recommended that there is at least 20 strains in each phenotype class assuming that the strains are not clonal. Additionally, aurora can be highly computationally demanding, with run-time largely dependent on the number of strains (Additional file 1: Fig. S7). Finally, *aurora* may struggle to identify the true autochthonous population if it is considerably smaller than the population of non-adapted strains. Therefore, we emphasize the necessity of validation experiments or the use of validation datasets to confirm the existence of causal links between the identified genotype and the analyzed phenotype. Users of any mGWAS tool should exercise caution, especially in the identification of low effect size variants, as the choice of algorithm, significance threshold, or prefiltering options can significantly influence the analysis outcome. It is noteworthy that the mGWAS tools employed herein produced divergent results despite utilizing the same input data. *aurora* consistently exhibited strong performance across various trait distributions and outperformed existing mGWAS tools when applied to diverse phenotypes such as invasiveness, virulence, antibiotic resistance, and host adaptation. *aurora* is an

accessible tool for entry-level users requiring minimal coding experience but modular enough to accommodate advanced analyses.

## Methods

### Input into *aurora*

*aurora* package contains two primary functions aurora_pheno() which identifies mislabeled strains and determines if the analyzed species harbors variants associated with the recorded trait and aurora_GWAS() which removes the mislabeled strains and calculates genotype–phenotype association scores. The required data inputs into aurora_pheno() are (i) a reconstructed phylogenetic tree imported as a phylo object or a pairwise distance matrix (kinship matrix) that represents the phylogenetic distances (i.e., the amino acid identity of conserved genes (cAAI) or average nucleotide identity (ANI) converted to distances, Mash distances [103], etc.). We recommend constructing the tree by aligning sequences of core genes using MAFFT [104] or MUSCLE [105]. This step is implemented in both the Panaroo [106] and Roary [107]. The aligned sequences can then be used as input into either IQ-TREE [108] or RAxML [109]. (ii) A pangenome matrix obtained from either Panaroo [106] or Roary [107] or any kind of presence/absence binary matrix (SNPs, *k*-mers, unitigs, etc.). *aurora* can also work with output from DRAM [110] which represents each strain as a set of modules and pathways. (iii) Lastly, a data frame that contains unique index for each strain and the corresponding trait value. Only categorical traits can be analyzed. The required input into aurora_GWAS() function is the same as for aurora_pheno() but if the results from aurora_pheno() are provided as well then all mislabeled strains are removed prior to the GWAS analysis. The workflow of *aurora* is depicted in Fig. 1.

### Implementation

This section contains a summary of the *aurora* package. Additional file 1 provides a more detailed description of each step with an example dataset. The first step removes features that have very low or very high presence frequency in the binary matrix. By default, features that are present in less than 3% (low_perc_cutoff) of strains in the analyzed dataset are removed. Features that are present in more than 99% (upp_perc_cutoff) of the strains are also removed. The user has the option to run the $\chi^2$ test with each feature (run_chisq). If the *p*-value of the $\chi^2$ test is high ($>0.1$), which means that the feature is uniformly distributed in the trait classes, then the feature is removed. This option is recommended only if the number of features is high after the initial frequency filter ($\sim 10,000$). The user also has the option to run an ancestral state filter (ancest_rec_filter). This step utilizes ancestral state reconstruction to identify genomic variants that are prevalent and highly mobile. These features are often fragments of larger gene families or associated with transposases, IS elements, or common plasmid elements (Additional file 2). Such features are not causal and should thus be removed.

The next step is to group strongly correlated features. Our package implements two methods for grouping. Firstly, a jaccard_filter in which a Jaccard distance is calculated between the features and the resulting distance matrix is used as an input to the DBSCAN algorithm [111]. The features that end up in the same cluster are then grouped. Secondly, the other method uses Hamming distance. The hamming_filter groups only

features that differ in the presence/absence in only $x$ strains where $x$ is a cut-off specified by the user (default = 3). Because DBSCAN has two parameters that usually require fine-tuning, the hamming_filter is simpler and thus a default option. If the dataset contains a large number of features that need to be collapsed, the jaccard_filter is then preferred because the hamming_filter collapses rare features with higher frequency which may result in a significant loss of information.

In the next optional step, the phylogenetic distances of outlying strains to the rest of the population are reduced (parameter reduce_outlier). If such outliers are present in the dataset, they will be sampled with higher frequency than the rest of the strains. This is usually desirable but if these deviations arose due to contamination or incompleteness of the genome assemblies, then such faulty assemblies may dominate the resulting training datasets. First, $z$-scores are calculated for either the tree branches or the pairwise distances in the input distance matrix. Pairwise distances or branch lengths higher than a threshold $\mu + 3 \bullet \sigma$ are assigned a value equal to the threshold. Elements of the pairwise distance matrix or branches of a tree that are equal to zero are assigned a new value that is equal to the minimal non-zero element present in the matrix or tree respectively.

Either Random walk algorithm or Phylogenetic walk algorithm is then applied to build a training dataset that captures the population structure of the species (default phylogenetic_walk). These algorithms are described in detail in Additional file 1. The algorithms either oversample or undersample the analyzed strains to yield a new dataset. A Phylogenetic walk produces training datasets that are accurate representations of the population structure while a Random walk only captures trends in the phylogenetic reconstruction. Both algorithms are stochastic and thus need to be repeated multiple times to ensure representative results are obtained. The number of repetitions is controlled by the parameter no_rounds. The default value is 100 but it can be reduced to lower the computational time.

### Threshold calculation phase

The goal of the next step is to construct threshold distributions and to find out if the species has genomic variants associated with the user-supplied phenotype. This is done by first intentionally mislabeling one strain from each class of the phenotype to all other classes and then calculating classification probabilities in the new and the original class. This process is repeated multiple times (specified by the parameter no_rounds). Each round four machine learning classification algorithms (random forest, AdaBoost, log regression, and classification and regression tree—CART) are used to get the classification probabilities. At the end of the Threshold Calculation Phase if the non-mislabeled strains belonging to class $x$ of the analyzed phenotype have significantly higher class $x$ classification probabilities than strains mislabeled into the class $x$, then the species is adapted to class $x$ of the phenotype. The Kolmogorov–Smirnov test is used to assess whether the differences are statistically significant. The output of the Threshold Calculation Phase is a square matrix containing $p$-values of the Kolmogorov–Smirnov tests. If both pairwise $p$-values of any two classes are above 0.05, then the classes are considered indistinguishable, and the species is not differentially adapted towards the two classes (see an example of the $p$-value matrix in Additional file 1: Fig. S1 and S2). Additionally, *aurora* in this step obtains distributions of classification probabilities of strains that were

intentionally mislabeled and strains that were not mislabeled. These distributions are used in the next phase to identify mislabeled strains.

### Outlier calculation phase

In this phase, *aurora* identifies strains that are mislabeled in the original dataset and strains that have low classification probability in their observed class but are not mislabeled (non-typical strains or weakly autochthonous strain in cases where the phenotype is habitat adaptation). Weakly autochthonous strains can be autochthonous in the observed class, but they are not typical representatives of the class. The process is the same as in the Threshold Calculation Phase, only this time none of the strains are intentionally mislabeled. First, the original dataset (without any intentional mislabeling) is resampled by either Random walk or Phylogenetic walk and then the resulting dataset is used for training the four machine learning models. The models are then used to predict class probabilities of the original dataset. This step is repeated multiple times (default: 100 times) and each time the classification probabilities of each strain are recorded. The results after finishing all the cycles of the Outlier Calculation Phase are sets of classification probabilities for each strain in each class. After sets of classification probabilities for each strain are obtained, the sets are compared to the sets of classification probabilities obtained in the Threshold Calculation Phase. The Kolmogorov–Smirnov test is used to calculate if the differences are significant. Additional file 1 contains a set of rules by which the strains' predicted class is determined. If the predicted class is the same as the observed, then the strain is considered autochthonous.

The output of the function aurora_pheno() is a list with multiple objects. For each machine learning algorithm, aurora_pheno() outputs a set of AUC (area under receiver operating characteristic curve) values documenting the classification performance of the tool. A set of feature importances of each machine learning tool is also generated. If parameter fitting was requested, then the fitted parameters are also part of the output. The main output objects are tables that detail the predicted class labels of each strain and three heatmaps that visualize the similarities between the strains. Two heatmaps are calculated based on the similarities derived from random forest models. The third heatmap is calculated based on the distances derived from CART models. It is essential to carefully examine these heatmaps to identify more potentially mislabeled strains. Examining these distance matrices can also reveal if there are multiple adaptation strategies to the phenotype classes.

### GWAS analysis

The GWAS analysis is governed by the function aurora_GWAS(). The input is the same as for aurora_pheno() and the function can be run even without a prior execution of function aurora_pheno(). The first (and optional) step of aurora_GWAS() is a reduction of the phylogenetic distances of outlying strains to the rest of the population (parameter reduce_outlier). This step is the same as in aurora_pheno(). In the following step, if the results from aurora_pheno() are available then all strains whose predicted class does not match the observed are removed (mislabeled strains). We refer to such strains as "mislabeled" further in this text. Parameter rm_non_typical controls if non-typical strains should be viewed as mislabeled (default: FALSE). aurora_GWAS() operates in

two modes: strict or consensus. The former removes a strain if it was classified as mislabeled by at least one machine learning model, and the latter removes a strain only if it was classified as mislabeled by all machine learning models. If the results from aurora_pheno() are not available, then no strain is removed. In the next phase, the dataset is resampled using Phylogenetic walk to capture the population structure. Then $\chi^2$ test with each variant is performed and the standardized residuals are computed from this test. Standardized residuals are a way to quantify how much the observed data differs from the expected data. Large positive standardized residual indicates a strong positive association with the class and vice versa. *aurora* also calculates precision, recall, and F1 value for each feature and each class treating each feature as if it was a decision stump. The output of aurora_GWAS() is a list with multiple objects. The main object is a data frame that shows F1 values and standardized residuals for each feature. The output object can also serve as input into other functions in the *aurora* package that execute other mGWAS tools with the filtered dataset.

### Simulated datasets

To test the performance of *aurora*, four different methods were used to simulate mGWAS datasets. The script used to benchmark Scoary [22] Simulate_pan_genome.py available here: https://github.com/AdmiralenOla/Simulate_pan_genome was used to simulate 200 strains with the causal gene penetrance set to 0.85 and 0.6. A phylogenetic tree was constructed by calculating pairwise Jaccard distance between all strains using the presence/absence matrix generated by the script and then a UPGMA algorithm was used to construct the tree. Only genes that were present in at least 75% of all strains were used for the Jaccard distance calculation. A pangenome simulation tool Simurg [33] was used to simulate a pangenome with 100 core genes and 200 strains. The file pan_genome_reference.fa from the pangenome construction of *Limosilactobacillus reuteri* was used as a reference multi-fasta file for the simulation. All other parameters were left at the default values. The resulting coalescent phylogenetic tree clearly showed two lineages (Fig. 2B). These two lineages have been assigned their unique phenotype class. The two multiple-state speciation and extinction models (MuSSE) were constructed using the tree.musse() function from diversitree R package [32]. The speciation and extinction rates as well as the transition matrices are shown in Fig. 2A. Two hundred strains were constructed in each MuSSE simulation. The tree.musse() function simulates the phylogeny and the distribution of the trait and its causal variants. To simulate the pangenome matrix containing the non-causal background genes, a function simulate_pan_panstripe() from the panstripe R package [112] was used. Panstripe uses a function simSeq() from package phangorn [113] to simulate the genes. The phylogenetic trees and the trait distributions with causal variants are shown in Fig. 2B. The simulations shown in Fig. 3A were each conducted one time. This was deemed sufficient because in our preliminary test, the variance of the results was small and independent of the tool and of how many strains were mislabeled.

### Real datasets

Available *Limosilactobacillus reuteri* genome assemblies were downloaded from NCBI in January 2023. Only assemblies that were between 1.8 and 2.4 Mb in size were retained,

and only assemblies with less than 200 contigs and confirmed NCBI taxonomic status were used. All human, poultry, porcine, rodent, and primate isolates (based on BioSample entries) were reannotated by PROKKA v1.14.6 [114] and the pangenome was then constructed with Panaroo v1.2.9 [106]. Panaroo was used with strict mode, and with parameters –merge_paralogs, –remove-invalid-genes, –remove_by_consensus. The phylogenetic tree was constructed by first aligning protein sequences of all single-copy core genes using MAFFT v7.490 [104] with a maximum number of iterative refinements set to 1000 and with 6merpair algorithm. The alignments were trimmed by trimAI v1.4.15 [115] with the default parameters. These alignments served as an input into IQ-TREE v2.2.0 [108]. IQ-TREE was run with the LG + G + F model and 1000 bootstrap replicates using UFBoot [116]. The tree was rooted by midpoint rooting. *Lactiplantibacillus plantarum* was used as a test case for a generalist species, and its genome dataset was constructed similarly. All available assemblies were downloaded from NCBI in January 2023. Only assemblies with lengths between 3 and 3.5 Mb, with less than 200 contigs and confirmed NCBI taxonomic status, were used. The pangenome and phylogenetic tree were constructed the same way as described above. To investigate pathogen genomes with low mutation and recombination rates, *Mycobacterium avium* subsp*. paratuberculosis* data were obtained by downloading all *Mycobacterium avium* assemblies from EnteroBase in February 2023 [117]. The pangenome was constructed as described above and diagnostic genes described in Bannantine et al. [118] were used to identify *M. avium* subsp. *paratuberculosis* (MAP) strains*.* Again, only strains with less than 200 contigs were used for the GWAS analysis. A *Salmonella enterica* serovar Typhimurium dataset was constructed similarly. First, 6039 genome assemblies with a contig count of less than 200 were downloaded from EnteroBase [117] and reannotated by PROKKA. This dataset was further narrowed by filtering genomes larger than 5.3 Mb and smaller than 4.7 Mb. Then the dataset was then split. The final 1223 genome assemblies were used to construct a pangenome as described above and the remaining 4816 genomes were used as a validation dataset. An *Escherichia* dataset as well as the associated experimental measurements were obtained from Galardini et al. [26], and the data were processed as described therein. Both *Neisseria meningitidis* datasets and the corresponding resistance and virulence metadata were obtained from the Neisseria Bacterial Isolate Genome Sequence Database (BIGSdb accessible at https://pubmlst.org/organisms/neisseria-spp, [119]). Only isolates from serogroup B were selected. Pangenome and phylogenetic tree were constructed as described above. The core genome SNPs for *Neisseria meningitidis* analysis were called using Snippy v4.6.0 (https://github.com/tseemann/snippy) with the genome assemblies as an input and strain 35,304 (assembly with the smallest number of contigs) as a reference. The unitigs were generated using unitig caller v1.3.0 (https://github.com/bacpop/unitig-caller), and Phandango [120] was used to visualize the results.

## GWAS analyses

All mGWAS tools including *aurora* use frequency filter to discard non-causal variants before the analysis and some also use *p*-value cutoffs after the analysis. These cutoffs were kept as low as possible so the maximum number of variants are reported and mean rank can be computed. When the simulated datasets were analyzed by *aurora*

the default parameters were used and only CART and random forest were run. *aurora* analysis of the *L. reuteri* dataset was run with default parameters and with all four machine learning algorithms. *L. plantarum* dataset analysis was run with all default parameters except no_rounds which was increased to 200, upp_perc_cutoff was changed to 95 and low_perc_cutoff was changed to 1. Because MAP is a species with low recombination and mutation rate *aurora* was run with cutoff_asr set to 3, low_perc_cutoff set to 1, and hamming_cutoff set to 1. *S.* Typhimurium analysis was run only with CART and random forest and with the following parameters: upp_perc_cutoff = 95, low_perc_cutoff = 5, hamming_cutoff = 5. The analysis of the two *Neisseria meningitidis* datasets with pangenome matrix as the input and the *Escherichia* dataset were all run with the default parameters. When penicillin resistance was analyzed with SNPs as the feature input, then the following parameters were used: low_perc_cutoff = 5, upp_perc_cutoff = 95, run_chisq = TRUE. Scoary version v1.6.16 [22] in all cases was run with 1000 permutations (-e 1000) and *p*_value_cutoff set to 0.99 so the order of all genes could be analyzed. The fixed effect model in Pyseer version v1.3.9 [21] was used as follows: First, the pairwise patristic distances were extracted from each phylogenetic tree by running Pyseer script phylogeny_distance.py. The resulting multidimensional scaling components from the kinship matrix were visualized in a scree plot and the number of components was chosen to preserve maximum information. The linear mixed models in Pyseer were constructed as shown in the Pyseer documentation (https://pyseer.readthedocs.io/en/master/). The elastic net models (enet) were constructed using the –wg enet flag. The mixing parameter ($\alpha$) was kept at the default value of 0.0069 which indicates low regularization. No sequence reweighting was carried out. Variants with $\beta$ (slope) lower than 0 were filtered out from the results of fixed effects models and the linear mixed models. TreeWAS [23] was run with default parameters except for *p*-value which was set to 0.99 to analyze the order of all features and n.snps.sim which was set to $100 \times$ the number of input features. All analyses with Hogwash [14] were also performed with the default parameters. Only fdr cutoff was set to 0.99 and the permutation number was set to 5000. When a mean rank was calculated the following function in R was used: base::rank(..., ties. method = "average").

### Gene annotation and analysis

Three tools were used to annotate the genes: eggNOG-mapper v2.1.6 [121], Kofam-Scan (https://github.com/takaram/kofam_scan) a CLI version of KofamKOALA [122], and a COG database [123]. In both COG and eggnog-mapper, diamond [124] in ultra-sensitive mode was used as the alignment tool. KofamScan on the other hand was used with HMMER v3.1 [125]. The parameter –tax_scope_mode in egg-NOG-mapper was set to "Bacteria". The result of the annotation is a.xlsx file that contains one sheet for each annotation tool. All these sheets contain annotation output from the tools for each gene but also confidence values. These are present because the annotation output represents the annotation of multiple sequences. The confidence values are the percentages of sequences that had this annotation. The COG and KofamScan sheets contain three confidence values. This is because each sequence can be associated with multiple COGs or KEGG identifiers respectively.

These COGs or KEGG identifiers are ordered based on their frequency, e.g., the most common identifier will be first. If the first confidence value is 80, it means that 80% of the sequences had this annotation as the most likely. If the second confidence value is 70, it means that 70% of the sequences had this annotation as the second most likely. iTOL [126] was used to visualize the phylogenetic trees. To verify the presence of high-ranking gene families from the GWAS analyses in the validation dataset of *S.* Typhimurium, HMMER v3.1 [125] was used. First, all the protein sequences of the gene were aligned using MAFFT as described above. Then the HMMR logo was built, and the validation dataset was searched with this logo as a query. The threshold for homology was an E-value $\leq 0.01$ and a score above 100. Unless specified otherwise, the threshold for statistical significance was $p < 0.05$, and only positive genotype–phenotype associations were evaluated.

## Supplementary Information

Additional file 1: Supplementary text and figures – Detailed explanation of aurora algorithm. Additional results and discussion [127–156].

Additional file 2: Ancestral reconstruction filter – All 119 Limosilactobacillus reuteri gene families that ancestral reconstruction filter removed during aurora run.

Additional file 3: MAP colonization factors – Annotated top Mycobacterium avium subsp. paratuberculosis colonization factors of bovine and ovine host.

Additional file 4: Typhimurium colonization factors – Annotated top 100 colonization factors of Salmonella enterica serovar Typhimurium clusters.

Additional file 5: Neisseria invasiveness – Results of aurora_GWAS() run with invasiveness phenotype and Neisseria meningitidis.

Additional file 6: Escherichia extraintestinal – Annotated top gene families associated with extraintestinal colonization humans by Escherichia sp.

Additional file 7: Lactiplantibacillus plantarum colonization factors – Annotated top 50 Lactiplantibacillus plantarum colonization factors for each habitat.

Additional file 8: Limosilactobacillus reuteri colonization factors – Annotated top 50 Limosilactobacillus reuteri colonization factors for each habitat.

Additional file 9: Peer review history.

### Acknowledgements
Not applicable.

### Review history
The review history is available as Additional file 9.

### Peer review information

### Authors' contributions
DB conceptualized the study, performed the data analysis, developed the software, and wrote the manuscript. JW and POT contributed to the manuscript preparation and provided critical revisions. All authors read and approved the final manuscript.

### Funding
This work was supported by Science Foundation Ireland through a Centre Award to APC Microbiome Ireland (12/RC/2273_P2). J.W. acknowledges support through an SFI Professorship (19/RP/6853).

### Data availability
The datasets supporting the conclusions of this article are available in the Zenodo repository DOI: 10.5281/zenodo.10397646 and additional files. *aurora* is an open-source package distributed under GPL-3.0 license. The compilation and installation can be done using R:
install.packages("*aurora*") # install aurora.
library(*aurora*) # attach aurora.

Some parts of **aurora** require Python3 and setting up a conda environment. Detailed vignettes with comprehensive tutorials are available here (https://dalimilbujdos.github.io/aurora/index.html). The tutorials demonstrate typical use cases, how to prepare the input objects, and how to post-process and interpret the output.

## Declarations

**Ethics approval and consent to participate**
Not applicable.

**Consent for publication**
Not applicable.

**Competing interests**
Not applicable.

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

## 