## [Additional file 9: Peer review history. · Genome Biology]

Review history

First round of review

Reviewer 1

This paper presents a new software, *aurora*, a tool for microbial GWAS. It is particularly aimed towards the handling of two problems. (1) Identifying features responsible for habitat adaptation, i.e. features that would be under strong selection and instrumental in shaping the genealogy of the microbes. This paper refers to them as "lineage effects", as a contrast to "locus effects", which have minimal effect on the overall population structure. (2) The handling of mislabeled and autochthonous strains (strains that have not - yet - adapted to their habitat. The paper is well written, but it is far too long. The main manuscript is a whopping 77 pages, and the supplementary adds another 55. I have seen master theses that are shorter than this, and I do believe it could be heavily trimmed down. A large part of the paper is devoted to the testing of *aurora* on different data sets, and I don't think all of these are needed as they largely make the same points and employ the same methodology. Perhaps all this information could be better presented in a "readthedocs" type format on the tool's webpage.

The tests in the manuscript also include benchmarking against other tools. Since a major part of *aurora* is the weeding out of "mislabeled" strains, the tests unsurprisingly show *aurora* as the best tool to use on a dataset where the authors have intentionally mislabeled a lot of strains. The manuscript makes a point out of the software not relying on p-values, instead reporting (so-called "directly interpretable") F1 scores and standardized residuals from the chi-square test. I do think this is slightly problematic because there is no a priori cutoff threshold and no statistics, the tool simply reports features by descending F1 score. This allows the authors to do some interesting post-hoc comparison for success criteria between the different benchmarked tools. For example "None of the mGWAS tools (Scoary, PySeer, HogWash, TreeWas) were able to identify these genes as significant cattle colonization factors. *Aurora* on the other hand ranked 8 of these 11 colonization factors in the top 84 genes." This seems a very arbitrary way to compare the success of different tools, and it looks like these other tools were run with default parameters and cutoffs set in a way that would advantage the no-cutoff software *aurora*.

As a side note, the manuscript claims that it is weird that these tools all output different p-values since they all test the same hypothesis - no relation between the feature and the phenotype. That's not weird at all, since the p-values are intrinsically tied to the specific statistical test, underlying assumptions, the input dataset after filtering (different in different tools) and for most tools even stochastic elements like bootstrapping. As a final note about this, I am not convinced that standardized residuals are directly interpretable OR that they are a better metric for association strength than p-values - The two should be highly correlated. The manuscript talks a lot about the number of false positives output by other tools even when the input phenotype is randomly assigned, but neglect to mention that these numbers are expected when cutting off at the $p=0.05$ level and you have thousands of features, or that their own tool can't have any false positives since there is no a priori definition for a positive.

As for the software itself I have not been able to test it, since a link has not been provided in the manuscript. I do think this is an absolute must for a method paper, and I can't properly review the paper without it. The methods seem interesting, and I am particularly interested in the `aurora_pheno()` function that is essentially a type of outlier analysis. The `aurora_gwas()` function

seems more straightforward, and, to my understanding, doesn't perform any correction for phylogenetic relationship other than using the relationship to create bootstrapped datasets. It seems to be like this would not lend itself well to the analysis of "locus effects", but without having tested the software it is impossible for me to tell.

Reviewer 2

Bujdos and colleagues present a novel microbial genome-wide association study (mGWAS) tool called aurora. The primary novelty of the approach is its attempt to generalize mGWAS to detect both lineage and locus contributions to phenotypic variation. To accomplish this, aurora performs two steps. First, the pheno tool is employed to evaluate the underlying population structure of the data, and identify potential misclassified samples. For former is a nice feature in identifying distinct sub-classes in the data, and the latter is emphasized as critical to mGWAS in the context of niche adaptation, where transient associations can confound mGWAS. The second step in aurora is performing a GWAS, wherein the approach to control for population structure is to sample the genetic diversity for each phenotypic class using one of two algorithms that attempt to decrease sampling of over-represented lineages. After describing the approach, aurora is evaluated in a variety of contexts, showing its strong performance against other common mGWAS tools using both simulated and real data.

Overall, this is a nice contribution to the mGWAS field. The rationale for the approach is sound, and it is evaluated thoroughly. I have some critiques to enhance interpretability of some of the results, as well as some comments based on trying to use the software.

Major comments

1. For the simulations in Figure 1, it's unclear how many trials were run for each case, which should be described in the legend. A few other concerns with this plot. First, it is indicated that if a causal locus was not returned by a method then a rank of 10,000 was used. It seems this could have a major impact on the mean rank, and perhaps not enable a true understanding of performance. Ideally the distributions underlying those bar plots showing the ranks of causal loci, along with a mean/median marker would be shown. A second thing that is unclear is how ties are handled for ranking. For instance, with the permutation-based tests it is possible for multiple genes to have the same minimum p-value.

2. Something that isn't communicated at all but is practically very important are run-time characteristics. Memory and runtime for data sets of different sizes should be reported, and a sense provided to the user what are critical determinants of performance (i.e. number of genomes, number of variants, phylogenetic structure, etc.) and what are the scale of studies that can be feasibly undertaken with aurora. Using the simulated data from Figure 1 seems good for this sort of evaluation.

3. Not much time was spent on the actual GWAS approach, which is in essence two alternative algorithms for sampling isolates to control for population structure. For the random walk algorithm described in the supplement, it seems that it could be highly susceptible to getting stuck in in parts of the tree (i.e. bouncing back and forth between outlier clades/strains) - was this observed in practice? For the phylogenetic sampling, it wasn't clear what the tip weights ended up being based on the description in the supplement. Is it just the sum of branch lengths leading to a tip in a phenotypically pure sub-lineage?

4. It would be helpful to have a limitations section to help provide guidance of cases when perhaps aurora might not be a good choice. If there really is strong population structure (e.g. one lineage associated with a phenotype), is aurora just returning lineage defining variants/genes? If so, are these strong hypotheses? Conversely, what about a case where a phenotype emerges many times independently (e.g. antibiotic resistance)? Would aurora pheno be prone to throw about small sub-lineages (e.g. singletons), that may actually be real?

5. The results are quite long. I understand that different examples are chosen to highlight the versatility of the tool, but it might improve readability to move some to the supplement and/or tool wiki.

Comments based on running software

aurora Pheno:

1. Provide a list of expected output from the function. For each expected output file, provide a brief description of each column, including a definition of the metric.

2. The tools tested in the dataset vary in computational complexity and run-time. Did the authors perform benchmarking on the memory and time requirements to run `aurora::aurora_pheno` & `aurora::aurora_GWAS`?

3. The following comments exist for each of the standard output files:

Heatmaps: Add legends for the row/column annotations and the cell fills for each heatmap. One is unable to interpret the colors without legends.

Strain classification files: (i) Report raw data used to determine the TRUE/FALSE classifications for each outcome group (e.g., poultry) in the CART and random forest files. (ii) Provide a definition for "TRUE", "INCONCLUSIVE", and "FALSE" output in these files. What is the threshold for confidence?

CART gfs output: (i) *.ratio columns: Provide numeric representations of the *.ratio columns. Without representations of what X|Y means, it is hard to interpret the values. Providing numeric values, such as proportion of isolates in the class with the genotype would be useful. (ii) *vs.* columns: What do these values represent? It is unclear the range of these values or how to interpret them.

Random Forest gfs output: (i) *.ratio columns: Provide numeric representations of the *.ratio columns. Without representations of what X|Y means, it is hard to interpret the values. Providing numeric values, such as proportion of isolates in the class with the genotype would be useful. (ii) MeanDecreaseGini_median & MeanDecreaseAccuracy_median: What do the column names represent? Reporting them as mean_*_median is not intuitive. Consider having a definition for the values in the manual page.

4. While `aurora_pheno` & `aurora_GWAS` worked on the in-package datasets, implementing

these packages on a custom, in-lab binary genotypic matrix and phenotypic results was not possible. We encountered the following problems that made it challenging to test `aurora_pheno`:

`aurora_pheno` using a custom binary genotype matrix:

(i) Outlier Calculation Phase: 100 CART model build out of 100 The estimated time to finish the Outlier Calculation Phase is 0 hours and 0 minutes.

```
Error in stats::hclust(dist_matrix, method = "complete", ) :
```

```
NA/NaN/Inf in foreign function call (arg 10)
```

```
Calls: <Anonymous> -> <Anonymous>
```

(ii) It is unclear whether this originated from the pre-processing steps.

5. How does `aurora_pheno` deal with outcomes of low-frequency?

`Aurora GWAS`:

1. What phylogenetic tree and binary genotypic matrix is used when the package performs `hogwash` & `treeWAS`?

(i) Does the `aurora_GWAS` implementation use the raw tree and binary matrix that is imputed? OR, does it use a curated dataset that results from the function's pre-processing steps OR `aurora_pheno` function? (ii) It is unclear how these algorithms are implemented in the manuscript and package. (iii) This will influence our ability to interpret these package's results. Using the raw tree, binary genotypic matrix, and phenotypic variable would be a more informative representation of how `aurora_GWAS` compares to the defaults of these comparator approaches.

2. For both `aurora_pheno` & `aurora_GWAS`, consider renaming the input `"binary_mat"` to `"binary_pangenome_mat"` to better reflect the input being a matrix from `roary` or `pangenome`. As many will use custom binary matrices, users may overlook the `"custom_bin_mat"` option found in the middle of the function.

3. Comment on the layout GWAS output file:

(i) `gene_family`: While useful for pangenome analyses, consider reporting this as `"genotype"` or `"feature"` to make it amenable to the diverse feature sets that can be used with this program. (ii) `*ratio` columns: These columns are useful, but using the term `'ratio'` to define the frequency of a feature in the class is not clear. Consider reporting this as proportion of frequency.

Authors' response to reviewers

Summary of manuscript changes

- We have addressed the reviewers' comments. All changes to the manuscript are outlined below. The line numbers are referencing the tract files. Additionally, the abstract was modified to meet the 100-word limit.

- A new figure (Figure S7) was added to the supplementary data. This figure documents `aurora`'s run-time dependence on input objects.

- We modified the code as outlined below.
- The software aurora is now public and available at <https://github.com/DalimilBujdos/aurora>.
- We made tutorials publicly available at <https://dalimilbujdos.github.io/aurora/>.

Reviewer reports:

Reviewer #1:

Review of the manuscript "aurora: A machine learning GWAS tool for analyzing microbial habitat adaptation".

This paper presents a new software, aurora, a tool for microbial GWAS. It is particularly aimed towards the handling of two problems. (1) Identifying features responsible for habitat adaptation, i.e. features that would be under strong selection and instrumental in shaping the genealogy of the microbes. This paper refers to them as "lineage effects", as a contrast to "locus effects", which have minimal effect on the overall population structure. (2) The handling of mislabeled and autochthonous strains (strains that have not - yet - adapted to their habitat. The paper is well written, but it is far too long. The main manuscript is a whopping 77 pages, and the supplementary adds another 55. I have seen master theses that are shorter than this, and I do believe it could be heavily trimmed down. A large part of the paper is devoted to the testing of aurora on different data sets, and I don't think all of these are needed as they largely make the same points and employ the same methodology. Perhaps all this information could be better presented in a "readthedocs" type format on the tool's webpage.

RESPONSE: We sincerely thank the reviewer for their fair and, in our view, accurate assessment of our mGWAS tool, as well as for highlighting potential areas of improvement in the manuscript. We also greatly appreciate the time and effort invested in thoroughly reviewing the extensive material we provided.

While we agree that the manuscript is lengthy, we believe the length is justified. aurora is a completely novel mGWAS tool, not merely a derivative of previous mGWAS or human GWAS tools. Specifically, aurora removes strains that it classifies as mislabeled. We consider this controversial due to potential concerns over the statistical validity of a dataset in which part of the strains was removed. The lengthy validation tests in the manuscript show that removing these mislabeled strains does not artificially

inflate association metrics but instead enhances the reliability of the results.

Related to the issue of excessive length, we respectfully disagree with the statement that the datasets analyzed and the conclusions drawn are the same across sections. In lines 152–176, we detail the differences between the simulated datasets. Additionally, the real datasets (*M. avium*, *S. Typhimurium*, *L. reuteri*, *L. plantarum*) differ significantly in their relevant characteristics: the *M. avium* dataset has lineage effects with a few mislabeled strains that can be spotted on the phylogenetic tree, *S. Typhimurium* has locus effects with many mislabeled strains, *L. reuteri* shows a mix of lineage and locus effects with many mislabeled strains, and *L. plantarum* is a generalist. These differences are outlined at the beginning of each section, and we also added a brief summary of these differences (lines 775 – 780). While the manuscript has 77 pages, only 34 pages focus on the core content (Introduction, Results, Discussion). Given the scope of species and phenotypes that aurora can analyze, we believe these datasets represent the minimum necessary to demonstrate the tool's reliability to readers and that the length is appropriate.

The supplementary material provides a detailed explanation of the aurora algorithm with a concrete example. Understanding, expanding and critiquing aurora would be difficult without this text. We envision aurora as a platform that can be expanded to accommodate additional functions. While the main text provides sufficient information for most readers, the supplementary material is essential for developers looking to build on aurora's capabilities. Since "readthedocs" is not a permanent format and is more suited for user information and tutorials, we prefer to keep the in-depth explanation of the tool within the publication.

The tests in the manuscript also include benchmarking against other tools. Since a major part of aurora is the weeding out of "mislabeled" strains, the tests unsurprisingly show aurora as the best tool to use on a dataset where the authors have intentionally mislabeled a lot of strains.

RESPONSE: Indeed, this is correct. However, we intentionally mislabeled strains only in the section titled "aurora can identify causal variants in simulated data despite inclusion of incorrectly labeled strains" which uses only simulated data. This section serves two purposes: first, it demonstrates that when no strains are mislabeled, aurora

either outperforms or matches the performance of existing mGWAS tools; second, it shows that when strains are mislabeled, the performance of other mGWAS tools declines, while aurora's performance remains largely unaffected (Figure 3A). This section is crucial as it proves that aurora can detect and eliminate mislabeled strains, unlike other tools whose performance deteriorates. The outcomes of this section are communicated to the reader in lines 231–236. In summary, we don't think the ability of aurora to identify mislabeled strains is over-stated.

The manuscript makes a point out of the software not relying on p-values, instead reporting (so-called "directly interpretable") F1 scores and standardized residuals from the chi-square test.

RESPONSE: One of the main points of the manuscript is not about aurora itself, but rather how other tools handle p-values and the multiple testing problem. In lines 718–733, we argue that p-values adjusted for multiple tests can never be fully accurate. While existing mGWAS tools adjust p-values to account for the nonindependence of the gene variants analyzed, the methods used for correcting for the multiple testing problem assume that presence/absence patterns of variants are independent of each other. This assumption is clearly incorrect (e.g., the presence of a gene encoding an enzyme in a pathway is dependent on the presence of other genes in that pathway). As a result, adjusted p-values cannot be interpreted in their true sense but rather used as an association metric. Since the adjusted p-values reported by other tools lose their true meaning, we chose to use two more intuitive association metrics, whose interpretation remains consistent. We removed the words "directly interpretable" from line 690 as we recognise that direct interpretability of F1 values and standardized residuals can be debated. Additionally, we added more text clarifying the role of the two association metrics (691 – 695).

I do think this is slightly problematic because there is no a priori cutoff threshold and no statistics, the tool simply reports features by descending F1 score. This allows the authors to do some interesting post-hoc comparison for success criteria between the different benchmarked tools. For example "None of the mGWAS tools (Scoary, PySeer, HogWash, TreeWas) were able to identify these genes as significant cattle colonization factors. Aurora on the other hand ranked 8 of these 11 colonization factors

in the top 84 genes." This seems a very arbitrary way to compare the success of different tools, and it looks like these other tools were run with default parameters and cutoffs set in a way that would advantage the no-cutoff software aurora.

RESPONSE: In the manuscript, we repeatedly emphasize that the significance threshold used by other tools can lead to false results (Figure S3, Figure S4, Table 1). Instead of relying on a predetermined threshold, in Conclusion (lines 818 – 820) we recommend that users validate their findings with a separate dataset, as we did in the section “aurora was the only GWAS tool tested that could identify genes responsible for host adaptation in *Salmonella enterica* serovar Typhimurium,” or by using metagenomic data to verify mGWAS results. As the reviewer rightly points out, it would be unfair to criticize other tools for failing to classify genes as significant without offering an alternative. Throughout the manuscript, we consistently highlight that aurora’s results have better gene rankings compared to other tools (Figure 3, 4, 5 and Table 1). In the case that the reviewer mentions, rather than reporting gene ranks for each individual tool and habitat (which would be too lengthy), we used a validation dataset and compared the residual deviance of a multinomial log-linear model fitted using the top genes (Figure 5C). All the tested mGWAS tools were run with very permissive pre-filtering and no p-value cutoff (see lines 1029 – 1061) so they report as many variants as possible. For the foregoing reasons, we have opted respectfully to not change the v1 submission on this point.

As a side note, the manuscript claims that it is weird that these tools all output different p-values since they all test the same hypothesis - no relation between the feature and the phenotype. That's not weird at all, since the p-values are intrinsically tied to the specific statistical test, underlying assumptions, the input dataset after filtering (different in different tools) and for most tools even stochastic elements like bootstrapping.

RESPONSE: Indeed, p-values are closely tied to the specific assumptions underlying a statistical test. In mGWAS analysis, the assumptions are minimal: bacterial populations reproduce through binary fission, and there is some level of convergent evolution (aurora does not assume this as it can also report lineage effects). Additionally, these tools assume that the genotype has had sufficient evolutionary time to confer a fitness

advantage to the strains possessing it. However, these assumptions are consistent across all mGWAS tools, so we would expect similar outputs from them. We added a text clarifying our position (687 – 688).

Initial filtering should not affect the results, as the filters used were very permissive and primarily aimed at reducing computational time. Typically in mGWAS analyses, the most common (>95%) and rarest (<1%) features are removed, as these are unlikely to be causal and would likely have p-values close or equal to 1. Therefore, this step should not impact the ranking of causal features. We added a text clarifying this approach (lines 1030 – 1033). Bootstrapping should also not influence the results, as the tools were set to perform numerous bootstrap iterations (e.g., treeWAS = 100 × the number of input features, Hogwash = 5000) to ensure the reproducibility of the findings. The bootstrapping parameters are described in lines 1056 – 1060.

As a final note about this, I am not convinced that standardized residuals are directly interpretable OR that they are a better metric for association strength than p-values - The two should be highly correlated.

RESPONSE: The p-values generated by other tools and F1 values or standardized residuals produced by aurora are not correlated, as demonstrated in the manuscript (i.e., Table 1). Standardized residuals are correlated with the chi-square p-value only when the tested phenotype has two categories. The chi-square test assesses whether there is a significant association between categorical variables. When the phenotype has more than two categories, the standardized residual for one phenotype category could be large, while the p-value might be lower because the variable is distributed evenly in the remaining phenotype categories. We added a short note (lines 682 – 685) explaining this issue.

The manuscript talks a lot about the number of false positives output by other tools even when the input phenotype is randomly assigned, but neglect to mention that these numbers are expected when cutting off at the $p=0.05$ level and you have thousands of features, or that their own tool can't have any false positives since there is no a priori definition for a positive.

RESPONSE: Indeed, if p-values were evenly distributed, we would expect 5% of the tested variants to be false positives. However, as we now show in references on line

263, p-values from mGWAS tests are not evenly distributed. This makes it impossible to accurately predict how many false positives would occur under the null hypothesis. Therefore, it's ultimately up to the reader to determine whether the numbers presented in Figure 3C are justifiable. We added a text explaining this predicament in lines 261 – 264.

Even with multiple test adjustment methods applied, some mGWAS tools still report false positives, as shown in Figure S3. While aurora cannot produce false positives, the goal of this chapter is not to directly compare aurora with other tools or to establish its superiority. We have expanded the text to clarify this point (lines 267 – 268).

As for the software itself I have not been able to test it, since a link has not been provided in the manuscript. I do think this is an absolute must for a method paper, and I can't properly review the paper without it.

The methods seem interesting, and I am particularly interested in the `aurora_pheno()` function that is essentially a type of outlier analysis. The `aurora_gwas()` function seems more straightforward, and, to my understanding, doesn't perform any correction for phylogenetic relationship other than using the relationship to create bootstrapped datasets. It seems to be like this would not lend itself well to the analysis of "locus effects", but without having tested the software it is impossible for me to tell.

RESPONSE: We provided source code for aurora immediately when asked by the editor. aurora is now publicly available here: <https://github.com/DalimilBujdos/aurora>

Reviewer #2:

Bujdos and colleagues present a novel microbial genome-wide association study (mGWAS) tool called aurora. The primary novelty of the approach is its attempt to generalize mGWAS to detect both lineage and locus contributions to phenotypic variation. To accomplish this, aurora performs two steps. First, the pheno tool is employed to evaluate the underlying population structure of the data, and identify potential misclassified samples. For former is a nice feature in identifying distinct subclasses in the data, and the latter is emphasized as critical to mGWAS in the context of niche adaptation, where transient associations can confound mGWAS. The second step in aurora is performing a GWAS, wherein the approach to control for population

structure is to sample the genetic diversity for each phenotypic class using one of two algorithms that attempt to decrease sampling of over-represented lineages. After describing the approach, aurora is evaluated in a variety of contexts, showing its strong performance against other common mGWAS tools using both simulated and real data. Overall, this is a nice contribution to the mGWAS field. The rationale for the approach is sound, and it is evaluated thoroughly. I have some critiques to enhance interpretability of some of the results, as well as some comments based on trying to use the software.

RESPONSE: We appreciate the thorough evaluation of our software and the overall positive assessment of our work.

Major comments

1. For the simulations in Figure 1, it's unclear how many trials were run for each case, which should be described in the legend. A few other concerns with this plot. First, it is indicated that if a causal locus was not returned by a method then a rank of 10,000 was used. It seems this could have a major impact on the mean rank, and perhaps not enable a true understanding of performance. Ideally the distributions underlying those bar plots showing the ranks of causal loci, along with a mean/median marker would be shown. A second thing that is unclear is how ties are handled for ranking. For instance, with the permutation-based tests it is possible for multiple genes to have the same minimum p-value.

RESPONSE: We assume the Reviewer is referring to Figure 3, not Figure 1. In Figure 3, we ran only one trial for each example. We faced a choice: either provide statistically tested results, which would require increasing the number of trials to at least 10. This would be very time-consuming and would force us to limit the comparison in other ways, such as reducing the range of mislabeled strains, the number of simulated datasets, or the number of tools tested. Instead, we opted to run a single trial across a wide variety of conditions. This decision was based on preliminary testing, which showed that the variance of results across multiple trials was low and stable across tools. This is reasonable since the simulated datasets do not contain outliers and are large enough to ensure that the random picking of intentionally mislabeled strains is

representative. We have revised the text to make this clear to the reader (description of Figure 3: 1553 and Methods: 986 – 988).

Regarding the rank of 10,000 for a tool that does not report a causal variant, we acknowledge that this approach will skew the mean rank but this consequence was intentional. We assumed that if a tool did not report a causal variant, the variant is at the end of the list with a p-value equal to 1. The number 10,000 was chosen because, in our experience, it reflects the typical pangenome size of a medium-sized dataset (~500 strains). Except for Hogwash, this rule was rarely applied. We changed the text of the manuscript to clarify our position (189 – 192).

While reporting bar plots for each trial would help the reader understand each tool's performance in greater detail, we would need 180 such plots which would be overwhelming. Using median instead of mean, or reporting both, is also not ideal, as the median of ranks does not accurately convey the data's variance. If we were analyzing raw p-values, the median would be more appropriate since the data would not be normally distributed and will have large relative range. However, we are analyzing ranks of p-values, where the effect of outliers has already been minimized (ranking the values transforms continuous data into ordinal which is the first step of many non-parametric tests), making the mean a more suitable measure. For example, consider the following p-values of causal variants: 0×10^{-6} , 3.9×10^{-6} , 6.3×10^{-2} , 1.6×10^{-1} . The corresponding ranks are 1, 2, 228, 643. These values were taken from Subsequent test used on MuSSE2 model. The mean of these values is 218.5 while the median is 115. We argue that the mean is a superior measure of data center in this case because it is less dependent on small changes in central p-values. If the third rank was changed from 228 to 5 then the median would change to 3.5 which would inaccurately reflect the fact that p-value and the corresponding rank of the fourth causal variant was much higher than the rest. On the other hand, the mean would change to 162.75 which better reflects the data structure. When the mean is calculated, outliers can be a concern. However, by using ranks instead of raw values, the impact of outliers is minimized. We respectfully chose not to change the manuscript text or figures, as we believe our approach is the most appropriate.

We added a description in the manuscript of how rank ties were handled (1060 –

1061). In short, we used the average rank. Consider the case where we have four variants all occupying third place then the rank was calculated as $(3+4+5+6)$; all places that the variants are occupying divided by 4; the number of variants occupying the same place.

2. Something that isn't communicated at all but is practically very important are runtime characteristics. Memory and runtime for data sets of different sizes should be reported, and a sense provided to the user what are critical determinants of performance (i.e. number of genomes, number of variants, phylogenetic structure, etc.) and what are the scale of studies that can be feasibly undertaken with aurora. Using the simulated data from Figure 1 seems good for this sort of evaluation.

RESPONSE: This is an excellent point. We have added a new figure (Figure S7) to the supplementary data that details the run-time and memory usage under various conditions. For this test, we created a new random dataset that allows for better control over key parameters. Specifically, we examined how dataset properties such as the number of strains, phenotype categories, and variants affect run-time. Additionally, for future users looking to optimize aurora for large datasets, we have provided a vignette (https://dalimilbujdos.github.io/aurora/articles/panGWAS_with_large_dataset.html).

3. Not much time was spent on the actual GWAS approach, which is in essence two alternative algorithms for sampling isolates to control for population structure. For the random walk algorithm described in the supplement, it seems that it could be highly susceptible to getting stuck in in parts of the tree (i.e. bouncing back and forth between outlier clades/strains) - was this observed in practice?

RESPONSE: In theory, distant clades or strains should be overrepresented because they capture unsampled variability. However, in practice, large distances between strains are often the result of poor assembly quality or an incorrect model choice for phylogenetic reconstruction. Both random walk and phylogenetic walk approaches can lead to an overrepresentation of outliers in the training data or in `aurora_GWAS()`, as shown in Figures S14 and S15. To address this, we have implemented a method to reduce outliers in the data and effectively “shrink” the phylogenetic tree or kinship matrix. This method is controlled by the `reduce_outlier` argument and is explained on pages 16 and 17 of the supplementary material.

For the phylogenetic sampling, it wasn't clear what the tip weights ended up being based on the description in the supplement. Is it just the sum of branch lengths leading to a tip in a phenotypically pure sub-lineage?

RESPONSE: In the phylogenetic walk, the probability weights WL and WD described in Figure S13 refer indeed to the sum of branch lengths in each subtree. We further clarified this in the description of Figure S13 (page 22 of the Supplementary materials)

4. It would be helpful to have a limitations section to help provide guidance of cases when perhaps aurora might not be a good choice. If there really is strong population structure (e.g. one lineage associated with a phenotype), is aurora just returning lineage defining variants/genes? If so, are these strong hypotheses? Conversely, what about a case where a phenotype emerges many times independently (e.g. antibiotic resistance)? Would aurora pheno be prone to throw about small sub-lineages (e.g. singletons), that may actually be real?

RESPONSE: Despite unwillingness to add even more length to the manuscript, we have now added a brief description of aurora's limitations (809 – 818). As correctly noted, in cases of strong population structure, aurora will only identify lineage-specific variants. We demonstrated this in our analysis of *Mycobacterium avium* subspecies tuberculosis. Due to linkage disequilibrium, not all of these variants are causal for the phenotype. However, when the presence/absence patterns of causal and non-causal variants are the same, no statistical method can distinguish between them. This is why, in the Conclusion section, we emphasize the importance of using validation techniques to confirm GWAS results.

A limitations paragraph was thus added to the Conclusion section. We describe two main limitations of the aurora algorithm: First, aurora may fail to detect the true autochthonous population if it is too small compared to the population of non-adapted strains. Second, if the number of strains in phenotype categories is too unbalanced (e.g., 20:1), aurora may oversample the smaller category, leading to inflated association metrics. To address these issues, aurora now prints new warning messages if the number of strains in a category is too low or if the phenotype categories are too unbalanced.

5. The results are quite long. I understand that different examples are chosen to

highlight the versatility of the tool, but it might improve readability to move some to the supplement and/or tool wiki.

RESPONSE: Reviewer 1 also mentions the length of the manuscript being problematic and suggested moving parts of it to an online wiki format. We oppose this idea because the supplement contains vital information for anyone wishing to understand how aurora works and expanding on our software. Our tool has a wiki page (<https://dalimilbujdos.github.io/aurora/>) but the purpose of this is to introduce aurora to the average user, to show typical usage cases, and to describe the outputs. We believe that the current mGWAS literature lacks in-depth algorithm descriptions and that this information deficit has led to the current mGWAS tools not being expanded or further developed. For these reasons, we ask to retain the length of the v2 submission.

Comments based on running software

We would sincerely like to thank the reviewer for these excellent tips. We ended up implementing most of these suggestions and further expanded on some ideas, which was helpful.

aurora Pheno:

1. Provide a list of expected output from the function. For each expected output file, provide a brief description of each column, including a definition of the metric.

RESPONSE: We added a brief overview of the output objects and files to R man pages. We also created a wiki page that describes each output file in detail (<https://dalimilbujdos.github.io/aurora/articles/outputs.html>).

2. The tools tested in the dataset vary in computational complexity and run-time. Did the authors perform benchmarking on the memory and time requirements to run `aurora::aurora_pheno` & `aurora::aurora_GWAS`?

RESPONSE: Even on very large datasets `aurora_GWAS()` takes only a few minutes.

We thus did not evaluate the run-time and memory usage of this function.

`aurora_pheno()` on the other hand may take several hours to finish. We added a new Figure (Figure S7) that documents the run-time and memory requirements across multiple different datasets.

3. The following comments exist for each of the standard output files:

Heatmaps: Add legends for the row/column annotations and the cell fills for each

heatmap. One is unable to interpret the colors without legends.

RESPONSE: We added a legend to the three output heatmaps that explains the color coding in the rows and columns. Additionally, we included a scale to indicate the distribution of values across the heatmap. However, the exact values in the heatmaps have a complex meaning. Users should primarily focus on the overall structure of the heatmaps and the associated cladograms, rather than the specific values.

Strain classification files: i) Report raw data used to determine the TRUE/FALSE classifications for each outcome group (e.g., poultry) in the CART and random forest files. (ii) Provide a definition for "TRUE", "INCONCLUSIVE", and "FALSE" output in these files. What is the threshold for confidence?

RESPONSE: The process of determining the labels "TRUE", "INCONCLUSIVE", and "FALSE" is rather complicated and is described on pages 36 – 39 in the supplementary material. The number of values used to determine these labels grows exponentially with the number of phenotype categories analyzed. We think that reporting such large number of raw values will not enhance the interpretability of the results. The thresholds used to evaluate the logical statement that is later used to determine the label is $\alpha = 0.05$.

CART gfs output: (i) *.ratio columns: Provide numeric representations of the *.ratio columns. Without representations of what X|Y means, it is hard to interpret the values. Providing numeric values, such as proportion of isolates in the class with the genotype would be useful. (ii) *.vs.* columns: What do these values represent? It is unclear the range of these values or how to interpret them.

RESPONSE: As mentioned above we created a wiki page for aurora (<https://dalimilbujdos.github.io/aurora/articles/outputs.html>), where all the columns in the output files are thoroughly described. Our decision to report raw data instead of a single numeric ratio is intentional. The values X|Y are calculated from the original dataset before phylogenetic or random walk is applied. Here, X represents the number of strains where the variant is present, and Y represents the total number of strains in that phenotype category. Although working with the unadjusted dataset is generally not appropriate because of population structure, we include these values for the user's convenience. The presence frequency in the adjusted dataset is reported in the

aurora_GWAS output file. In this case, Y is constant across all phenotype categories, allowing us to report a single value.

In the CART output file, the *vs.* columns refer to the importance measures used in the respective models. If three phenotype categories are analyzed (e.g., A, B, C), three CART models are built in each round: A vs B, A vs C, and B vs C. The column values represent the sum of the goodness-of-split measures for each split in the tree, indicating how well each variant contributes to the model's performance. We have renamed these columns to “goodness_of_split_*vs*”.

Random Forest gfs output: (i) *.ratio columns: Provide numeric representations of the *.ratio columns. Without representations of what X|Y means, it is hard to interpret the values. Providing numeric values, such as proportion of isolates in the class with the genotype would be useful. (ii) MeanDecreaseGini_median & MeanDecreaseAccuracy_median: What do the column names represent? Reporting them as mean_*_median is not intuitive. Consider having a definition for the values in the manual page.

RESPONSE: The meaning of the *.ratio columns is the same as in the previous case.

MeanDecreaseGini_median and MeanDecreaseAccuracy_median both refer to the feature importance in the model. Mean Decrease Gini is a measure of a feature's importance based on the Gini impurity index used in decision trees.

MeanDecreaseGini_median is thus median of these values over all models constructed in the Outlier Calculation Phase. Mean Decrease Accuracy evaluates feature importance by measuring how much the model's accuracy decreases when the values of that feature are randomly shuffled. Similarly, MeanDecreaseAccuracy_median is a median of all these values calculated from the constructed random forest models.

4. While aurora_pheno & aurora_GWAS worked on the in-package datasets, implementing these packages on a custom, in-lab binary genotypic matrix and phenotypic results was not possible. We encountered the following problems that made it challenging to test auroa_pheno:

aurora_pheno using a custom binary genotype matrix:

(i) Outlier Calculation Phase: 100 CART model build out of 100 The estimated time to finish the Outlier Calculation Phase is 0 hours and 0 minutes.

Error in stats::hclust(dist_matrix, method = "complete",) :

NA/NaN/Inf in foreign function call (arg 10)

Calls: <Anonymous> -> <Anonymous>

(ii) It is unclear whether this originated from the pre-processing steps.

RESPONSE: Thank you for pointing this out to us. This issue has now been resolved.

5. How does aurora_pheno deal with outcomes of low-frequency?

RESPONSE: We are not sure what “outcomes of low-frequency” exactly means in this context.

Aurora GWAS:

1. What phylogenetic tree and binary genotypic matrix is used when the package performs hogwash & treeWAS?

(i) Does the aurora_GWAS implementation use the raw tree and binary matrix that is imputed? OR, does it use a curated dataset that results from the function's preprocessing steps OR aurora_pheno function? (ii) It is unclear how these algorithms are implemented in the manuscript and package. (iii) This will influence our ability to interpret these package's results. Using the raw tree, binary genotypic matrix, and phenotypic variable would be a more informative representation of how aurora_GWAS compares to the defaults of these comparator approaches.

RESPONSE: Which tree, binary and phenotype matrix is used for hogwash and treeWAS is dictated by what the input into aurora_GWAS() is. If the user supplies results from aurora_pheno() then the mislabeled strains are removed and then hogwash and treeWAS are run. If the results of aurora_pheno() are not provided then no strains are removed prior to running hogwash and treeWAS. Before running treeWAS all features that were not observed in any of the strains are removed. The user can modify the parameter n.snps.sim (default = 100*number of features) while other parameters are set to: p.value.correct = "fdr", plot.tree = FALSE, plot.manhattan = TRUE, plot.null.dist = TRUE, plot.dist = TRUE, chunk.size = 1000. In hogwash these parameters are used as default: test = "both", fdr = 0.99, perm = 5000 and the user can modify the perm parameter. The running parameters are described in “outputs” vignette.

2. For both aurora_pheno & aurora_GWAS, consider renaming the input

"binary_mat" to "binary_pangenome_mat" to better reflect the input being a matrix from roary or pangenome. As many will use custom binary matrices, users may overlook the "custom_bin_mat" option found in the middle of the function.

RESPONSE: Indeed, this type of inputting might be confusing and prone to potential errors. We thus decided to completely reformat the input arguments. Now the binary matrix irrespective of its origin is supplied via argument bin_mat. There is additional argument called type_bin_mat which specifies the origin of the binary matrix. The default is "roary" but alternatives are "panaroo", "DRAM", "custom", "SNPs", "SCARAP", "unitigs", "k-mers" and "PIRATE".

3. Comment on the layout GWAS output file:

(i) gene_family: While useful for pangenome analyses, consider reporting this as "genotype" or "feature" to make it amenable to the diverse feature sets that can be used with this program. (ii) *ratio columns: These columns are useful, but using the term 'ratio' to define the frequency of a feature in the class is not clear. Consider reporting this as proportion of frequency.

RESPONSE: We acknowledge that these terms could be confusing. The column gene_family was changed to "Variant" and the ratio column were renamed to "frequency" which should clarify.

Second round of review

Reviewer 1

The authors have done a commendable job of responding to all issues I raised in the previous round of review. I remain convinced that the paper is far too long, but in the end that will be up to the editors.

Reviewer 2

The authors have sufficiently addressed my comments.